**Surface formation, preservation, and history of low-porosity crusts at the**
**WAIS Divide site, West Antarctica.**
John M. Fegyveresi[1,2], Richard B. Alley[2], Atsuhiro Muto[3], Anaïs J. Orsi[4],
Matthew K. Spencer[5]
[1]Terrestrial and Cryospheric Sciences Branch, U.S. Cold Regions Research and
Engineering Laboratory (CRREL), Hanover, NH, 03755, USA.
[2]Dept. of Geosciences, and Earth and Environmental Systems Institute, Pennsylvania
State University, University Park, PA, 16802, USA.
[3]Dept. of Earth and Environmental Science, College of Science and Technology, Temple
University, Philadelphia, PA, 19122, USA.
[4]Laboratoire des Sciences du Climat et de l'Environnement, LSCE/IPSL, CEA-CNRS-
UVSQ, Université Paris-Saclay, F-91191, Gif-sur-Yvette, France.
[5]School of Physical Sciences, Lake Superior State University, Sault Sainte Marie, MI,
49783, USA.
*Correspondence to:*
J. M. Fegyveresi (fegy.john@gmail.com; john.m.fegyveresi@usace.army.mil)
**Key Words:**
• Antarctic snow surface, ice cores, field observations, snow-surface crusts, bubble-
free layers, vapor transport, firn properties, snow physics.

32                                    **Abstract**

Observations at the WAIS Divide site show that near-surface snow is strongly altered by

weather-related processes such as strong winds and temperature fluctuations, producing features
that are recognizable in the deep ice core. Prominent "glazed" surface crusts develop frequently at
the site during summer seasons. Surface, snow pit, and ice core observations made in this study
during summer field seasons from 2008-09 to 2012-13, supplemented by Automated Weather
Station (AWS) data with short and longwave radiation sensors, revealed that such crusts formed
during relatively low-wind, low-humidity, clear-sky periods with intense daytime sunshine. After
formation, such glazed surfaces typically developed cracks in a polygonal pattern likely from
thermal contraction at night. Cracking was commonest when several clear days occurred in
succession, and was generally followed by surface hoar growth; vapor escaping through the
cracks during sunny days may have contributed to the high humidity that favored nighttime
formation of surface hoar. Temperature and radiation observations show that daytime solar
heating often warmed the near-surface snow above the air temperature, contributing to upward
mass transfer, favoring crust formation from below, and then surface hoar formation. A simple
surface energy calculation supports this observation. Subsequent examination of the WDC06A
deep ice core revealed that crusts are preserved through the bubbly ice, and some occur in snow
accumulated during winters, although not as commonly as in summertime deposits. Although no
one has been on site to observe crust formation during winter, it may be favored by greater
wintertime wind-packing from stronger peak winds, high temperatures and steep temperature
gradients from rapid midwinter warmings reaching as high as -15°C, and perhaps longer intervals
of surface stability. Time-variations in crust occurrence in the core may provide paleoclimatic
information, although additional studies are required.  Discontinuity and cracking of crusts likely
explain why crusts do not produce significant anomalies in other paleoclimatic records.

**1: Introduction**

Visual and thin-section examination of the WAIS Divide deep ice core from West Antarctica revealed an annual signal linked to bubble and grain characteristics [Fitzpatrick et al., 2014], but also numerous crusts. These crusts are bubble-free or nearly so, typically one grain and 1 mm or less in thickness, and are readily identified visually in bubbly ice (Fig. 1). Their presence in greater abundance than seen in most cores [e.g., Alley, 1988] motivated studies to understand their formation, possible influence on other paleoclimatic data, and potential for recording paleoclimatic conditions themselves.

Work by Orsi et al. [2015] and Mitchell et al. [2015] showed that no significant artifacts are introduced to paleoclimatic records by the WAIS Divide crusts. Here, we report additional studies showing that summertime crusts form under specific conditions linked to persistent high-pressure systems, so the time-series of crusts likely contains paleoclimatic information; however, many additional issues must be addressed before useful climate histories could be constructed confidently.

Bubble-free layers much thicker than the bubble-free crusts discussed here are sometimes observed in ice cores from warm sites, and provide evidence of refrozen meltwater [e.g., Das and Alley, 2005]. These are of interest as paleoclimatic records but have the potential to anomalously distort records of trapped gases or other components of ice cores. Refrozen meltwater can be identified by an excess of trapped heavy noble gases, so Orsi et al. [2015] analyzed WAIS Divide samples containing bubble-free crusts, finding that not enough meltwater was involved to significantly perturb records of other trace gases. Additionally, crusts might greatly modify gas trapping in the firn, but measured nitrogen-isotopic ratios at WAIS Divide show that gravitational fractionation occurs down to the normal trapping depth where normal amounts of air are trapped,

demonstrating that the crusts are not both impermeable and laterally extensive at shallow depth
[Mitchell et al., 2015; Battle et al., 2011].

Here, we report coordinated observations of crust formation over five summers (2008-09

to 2012-13) at the WAIS Divide site, involving daily observations of surface evolution, shallow
snow-pit studies with a 2-m pit at least once per year, insolation measurements, and near-surface
temperature profiling, supplemented with data from an on-site automated weather station (AWS).
We find that crusts form most commonly in the summer (45% greater occurrence), but do also
form in winter. In summer, crust formation primarily results from the effects of strong diurnal
temperature cycling under clear-sky, low-wind, relatively warm conditions. Wintertime
observations are not available, but the physical understanding gained from our summertime data
suggests hypotheses for formation. Time-trends in the occurrence of summertime crusts in the
core may reveal changes in the frequency of the persistent high-pressure conditions that generate
crusts, although additional work will be required to quantify this.
**2: Methods**

The main methods used are described here. Additional details are provided in Fegyveresi

[2015]. The surface was observed continually by one of us (JF) during the five field seasons
extending from 2008-09 to 2012-13 (Table 1). During each austral summer, a back-lit snow pit
was also prepared and studied (five total pits). All pits were sited within 1 km radius of the
primary ice-core drilling facility, but avoided regions disturbed by camp operations or the "drift
tail" of enhanced accumulation downwind of the camp. Following prior practice [e.g., Benson,
1962; Koerner, 1971; Alley, 1988], each sampling site involved excavating a pair of ~2 m cubic
pits separated by a wall ~0.5 m thick, with one pit left open to supply back-light, and the other a
roofed observation pit. Features such as crusts and hoar layers were easily identifiable from the
observation pit on the back-lit wall (Fig. 2). Pit walls were observed, mapped, sampled, and
photographed (tripod-mounted > ¼ s exposures). Each pit was oriented so the prevailing wind
direction, approximately north-south, ran from right-to-left along the back-lit wall.

An automatic weather station (AWS) on site at WAIS Divide (named Kominko-Slade in

the University of Wisconsin AWS system; Lazzara et al. [2012]), collected data on temperature,
air pressure, wind, and humidity starting in the 2009-10 season (all dates and times are GMT).
Beginning in 2011-12, upward-facing and downward-facing shortwave Li-Cor LI200
pyranometers were added initially 1 m above the surface to measure incoming and outgoing
shortwave radiation (0.4-1.1 μm spectral response). Both sensors were newly calibrated and
mounted in a cosine-corrected head (for solar angles up to 80°), with typical operational errors in
daylight of ±3% (max ±5%). A Kipp-Zonen CNR2 net radiometer with upward- and downward-
facing pyranometers and pyrgeometers was added on an AWS mounting arm during the 2012-13
season, in order to measure both net short and longwave radiation. This instrumentation replaced
the previous Li-Cor instrumentation. The pyranometers operated with a spectral response of 0.3 –
2.8 μm, operational errors of ±3.5 %, and sensitivity of 15.21 μV $W^{-1}$ $m^{-2}$, while the pyrgeometers
operated with a spectral response of 4.5–45 μm, operational errors of ±5.6 %, and a sensitivity of
12.52 μV $W^{-1}$ $m^{-2}$ respectively; typical impedances were ~7 ohms. All AWS relative humidity
values reported here are expressed in terms of saturation vapor pressure over ice and corrected for
low-temperature offsets (see Anderson, 1994).

Also during the 2012-13 season, we calibrated and installed five PRD (platinum

resistance detector) strings in the upper 5 m of firn in a 2 km survey line extending approximately
upwind (grid-west, true-north) starting ~50 meters from the on-site AWS. The strings were
designed by one of us (AM) following the procedures in Muto et al. [2011]. Each sensor string
was 5 m long and consisted of 16 individual PRDs (HEL-700 series; ±0.03°C accuracy, ±0.18°C
total combined error, including data-logger error) with denser sampling in the shallower firn to
capture the greater variability there (see also Supplemental Table S2). Sensor calibration took
place over a 60-minute period using a constantly-stirred ice-bath method, and then the newly
calibrated sensors were deployed incrementally over a 10-day period starting Dec. 15[th].
Deployment boreholes were drilled using a 4 cm diameter hand-auger, and then back-filled once
strings were installed. Campbell logging equipment (CR1000 data logger and AM/16/32
Multiplexer) and 12V sealed lead-acid batteries were housed in a foam-insulated wooden box
beside each borehole and just below the surface. The first string was placed 50 m from the AWS,
and the other strings were placed upwind of it by 10, 100, 1000, and 2000 m (Supplemental Table
S3). Measurements were taken every minute over the survey interval. Each 12V battery was
swapped out weekly with newly charged replacements to ensure that the sensor strings were
continually recording. During each site visit, we took photographs, and noted local
meteorological and surface conditions. Each sensor string took approximately 24 hours to
equilibrate with the surrounding snow following installation due to the backfilling of the open
boreholes with surface snow.

We studied crusts in the ice core as well as in the near-surface. As described in

Fitzpatrick et al. [2014], the entire deep core and various associated shallower cores were
inspected visually during core processing lines at the US National Ice Core Laboratory, primarily
by one of us (MS), but with some intercomparisons from other observers. The core was observed
on a light table in a darkened booth, and key features were noted on meter-length log books. The
crusts were easily visible as thin, glassy, bubble-free or nearly bubble-free layers (e.g. Fig. 1).

Annual cycles are visible in the bubbly part of the core, arising from the tendency for

near-surface processes to generate coarse-grained, low-density layers including depth hoar in
summer [Fitzpatrick et al., 2014; Fegyveresi, 2015]. However, annual-layer dating of the ice core
using electrical conductivity (ECM, which is primarily controlled by ice chemistry) and soluble-
ion chemistry proved more accurate than dating with visible strata [Buizert et al., 2015; Sigl et al.,
2016; WAIS Divide Project Members, 2013]. Here, we estimate the season in which each crust
occurs by assigning each summertime peak in the WD2014 time scale to January 1of its year, and
then linearly interpolating; accumulation at the site is relatively evenly distributed through the
year, justifying this approximation [Banta et al., 2008; Fegyveresi, 2015, Fegyveresi et al., 2016].

**3: Observations**
**3-1: Near-surface observations**

We summarize key observations on crust formation here. Additional information, and

complete narrative descriptions of particular crust-forming episodes, are provided in Fegyveresi
[2015].

Glazed crusts were repeatedly observed to form on the snow surface (Figs. 3 and 4),

primarily during late-December and January, with an interval between formation events of
roughly one and two weeks (see Figs. 5-8). Crust formation often followed a storm or wind event,
and occurred during a time of higher atmospheric pressure, light winds, clear sky, strong
insolation, large diurnal temperature cycling, and low relative humidity.

As shown in Figure 9, the crusts were often internally complex. The upper few

millimeters of firn were anomalously high-density (> 400 kg m$^{-3}$) and fine-grained, and might be
termed a multi-grain crust. Within this, and especially at the top, were one or more lower-porosity
single-grain crusts. To an observer, light reflected off these crusts gave the appearance of a glaze
on the snow surface. (e.g. Fig. 4), [see also Orsi et al., 2015, their Fig. 5].

Typically, a glazed crust started as isolated sub-meter to few-meter patches on unshaded

regions of the snow surface or sastrugi, which were most consistanly exposed to sunlight, and
spaced tens of meters to more than 100 m apart. The spatial size and extent of glazed crust
patches varied considerably and were not measured directly, however no single observed crust
patch was greater than 100 m in length in any dimension. Over the first days of formation, glazed
crusts expanded to form a laterally extensive interconnected surface broken by isolated sub-meter
to few-meter unglazed patches on shaded faces of sastrugi. Glazed crusts were most continuous
where the surface was smoothest. Reconnaissance surveys extending a few kilometers from camp
showed that glazed-crust formation was consistent at least that far.

Within 2-3 days of formation, glazed features developed prominent polygonal cracks

with few-meter spacing (e.g. Fig. 4). It is likely that these cracks formed by thermal contraction
during nighttime cooling, which was driven by the large diurnal temperature swings observed at
the time (see below). We excavated some cracks, which could be traced downward from the
surface typically ~20-30 cm.

A pronounced hoar began forming within 24 hours of the onset of cracking of the glazed

crust in each case observed (e.g. Fig. 3). Measured relative humidity was notably higher during
hoar formation (see Figs. 5-8) than before, and sometimes (e.g., January 7[th], 2010) a fog
developed early in the time of hoar formation, providing a source of vapor to the surface hoar
from above. Surface glazing was not required for formation of such hoar layers, as one formed
quickly on December 30[th], 2009 during a very warm (> -10°) fog episode with elevated measured
relative humidity, but without prior formation of surface glaze.

Hoar layers that we observed during the field seasons were subsequently either buried,

destroyed by wind, or gradually sublimated away over 2-3 additional days. We observed strong
winds remove hoar layers, with a threshold of $\sim7$ m s$^{-1}$ ($\sim13$ knots). In one case, hoar removal
required somewhat lower speed when wind was directed orthogonal to the prevailing direction
and thus sastrugi orientation, similar to observations by Champollion et al. [2013] at Dome C,
East Antarctica.

No above-freezing temperatures were observed by the AWS, but on January 2, 2011, the

temperature reached a high of -2.8ºC (see Fig. 6; Supplemental Fig. S1). While no direct surface
melt was observed, some melt was noted along exposed, vertically cut wall faces near the ice-
core drilling facility (Supplemental Fig. S2). A prominent multi-grain crust was observed the next
year in snow pits, likely dates from that time, and shows features that are consistent with some
melting-refreezing having occurred (Supplemental Fig. S3).

The PRD strings document strong variations in subsurface temperature, following the air

temperatures as expected. During the cooling phases of diurnal cycles, air temperatures (AWS)
and near-surface snow temperatures (S0) dropped well below temperatures deeper in the firn
including the shallowest in-firn sensor (S1) at ~20 cm (Figs. 10 and 11), with the surface as much
as 3ºC colder than firn at 40 cm (S2) depth (e.g. Supplemental Fig. S4). This would have driven
upward mass flux from the firn towards the surface.  Such conditions often developed when
surface hoar was forming from fog, and thus likely with a downward as well as an upward vapor
source to the near-surface layer.

**3-2: Snow-pit observations**

Each of the five snow pits showed a clear annual cycle in the visual stratigraphy, but with

notable "noise". Depth hoars occurred primarily in summertime layers and into autumn, but with
occasional hoar layers in winter and spring layers. Crusts were also most common in summertime
and into autumn, but not restricted to those times. Similar to the observations made by Alley
[1988] at other sites in Antarctica, sequences of strata at WAIS Divide typically showed lateral
continuity over 2 m scales, although with some variation. Many graded beds were also present,
likely indicative of changes during a specific storm event or primarily before the next storm. This
was later confirmed on-site with accumulation stakes and measurements following specific large
storm events [see also Koffman et al., 2014; Criscitiello et al., 2014].

The snow pits from the 2008-09, 2009-10, and 2010-11 seasons at WAIS Divide were

mapped here in greatest detail, and meter-wide sub-swaths of their complete pit-wall maps are
shown in Figure 12. Complex stratigraphy and variations are clearly discernable, and illustrate the
variability within 1 km of each other at WAIS Divide in contiguous years. This is likely
indicative of the influence of complex processes of deposition and metamorphism, with frequent
occurrences of depositional and erosional features (sastrugi, whalebacks, wind scoops, hollows,
etc.).  We chose annual layers in the pit maps based upon visual inspection in the field,
subsequent examination of photographs of the pits, and overall trends in measured densities (see
e.g. Fig. 13).
We measured pit bulk densities using 100 cm$^3$ stainless-steel, box-type cutters [e.g.,
Conger and McClung, 2009] and a digital scale accurate to 1 gram. Density samples were taken
in all five concurrent seasons' pits in duplicate, at ~5 cm intervals, from the pit side-wall (so as
not to disturb the back-lit wall).  These duplicates were then averaged together for final values.
Samples measured in the 2008-09 pit were taken with regards to marked strata, and therefore at a
slightly higher frequency. Density measurements from pits of all five seasons yielded an average
density of 386.6 ± 3.2 kg m$^{-3}$ for the upper 2 meters of firn (Fig. 14), all with a nearly identical
linear trend-line slope of ~0.4 kg m$^{-3}$ cm$^{-1}$ with depth.
Seasonal interpretations of all five pits indicated an average of ~3.75 years of
accumulation recorded over the 2 meter depths, which yields an average of ~0.53 m a$^{-1}$ of
accumulation at the average pit snow-density.  Converted to water-equivalent, this becomes ~0.20
m a$^{-1}$$_{w.e.}$ (or ~0.22 m a$^{-1}$$_{ice}$).  These values agree closely with recently published values [WAIS
Divide Project Members, 2013; Banta et al., 2008; Burgener et al., 2013].
We documented obvious crusts and hoar layers for each snow pit. Most commonly, crusts
occurred just above depth hoars, but crusts were observed without hoar, and hoar without crust.
Both single-grain-thick (~1 mm) and multi-grain (≥4 mm) crusts were observed, with the
common association of single-grain crusts in and usually at the top of multi-grain crusts as noted
above. All crusts had densities estimated over 400 kg m$^{-3}$. Counting a multi-grain crust containing
a single-grain crust as one feature, the five 2-meter snow pits revealed an average of ~18.8 ± 2.5
(±1σ) total crusts, or approximately 5 crusts per year.
**3-3: Ice-core data**
In the bubbly ice included in our crust logging (120-577 m depth) in the WAIS Divide
core, 10,268 crusts were identified (Fig. 15). A few were discontinuous across the core, or
displayed at least a few pores extending through; others appeared largely or completely
continuous and impermeable at the scale of the core. Experience with independent observers
showed little or no error in crust identification. We cannot rule out the possibility that bubble
migration contributed to loss of some crusts in the deepest bubbly ice considered, but the crusts
continued to be clear and readily identifiable, so we do not believe that the trend to fewer crusts
in the deepest ice is an artifact. We cannot fully exclude the possibility that there is an
observational bias related to the drop in crust prevalence over the most recent ~250 years, as the
crusts are more difficult to discern in the shallow firn.
The seasonal distribution of the crusts is shown in Figure 16. Crusts occur year-round,
but are ~45% more frequent in summertime accumulation than in wintertime. Certainly, the
natural variability in seasonal distribution of snow accumulation and in the timing of peak
impurity input mean that details of the shape of the seasonal distribution of crust occurrence are
notably uncertain. However, given the high reliability of the annual-layer dating, and the multiple
indicators that agree well [Buizert et al., 2015; Sigl et al., 2016; WAIS Divide Project Members,
2013], "summer" versus "winter" or "nonsummer" is well-constrained.
Time-trends of seasonal crust occurrence are also shown in Supplemental Figure S5,
separating the largely sunless winter (May-August) from the sunny spring-summer-fall
(September-April, with at least 8 hours of sunlight per day). Both first increase and then decrease
slightly over the 2400-year record, but with a larger relative change in the sunlight season.


**4: Synopsis and Discussion**

Our observations confirm and extend prior work on this topic [see e.g. Anderson and

Benson, 1963]. Depositional processes and metamorphism primarily in the snow that comprises
the upper few centimeters of the firn, produce prominent layering.  Wintertime accumulation,
while notably variable, is more homogeneous than summertime deposits, with wind-packed
layers prominent in winter, and more-variable layers including crusts and hoar more common in
summer [e.g., Sorge, 1935; Benson, 1962, Gow, 1965; 1969; Weller, 1969; Colbeck, 1982;
Colbeck, 1983; Alley, 1988; Alley et al., 1997]. These features are altered during subsequent
burial and conversion to bubbly ice, but still produce recognizable features in the ice core that
allow identification of annual layers and crusts [e.g., Alley et al., 1997; Fitzpatrick et al., 2014].

Our observations at WAIS Divide show repeating events that generate the main features

of the summertime accumulation. In a typical event, a storm with strong winds brings snow
accumulation, followed by a high-pressure system bringing clear skies, greatly reduced winds,
initially low humidity, and strong diurnal variations in sunshine, air temperature, and net surface
energy-balance.

Early in this clear-sky interval, the wind-packed upper surface develops a millimeter-

thick glazed crust or possibly crusts in a few-millimeters-thick multi-grain crust. Strengthening of
crusts over one to a few days, is followed by polygonal cracking from contraction caused by
nighttime cooling. Vapor released through the cracks contributes to rising relative humidity, and
surface-hoar deposition in subsequent nights. At WAIS Divide, evolution of the crust-hoar
complex typically is truncated by arrival of another storm, which may remove or bury the hoar,
and typically buries the crusts below the level of fastest metamorphism, allowing them to be
preserved.

Not every aspect of a typical event is observed in each case. Crusts form and can be

buried by additional snowfall without growth of a surface hoar on top of them. Crusts are
somewhat discontinuous, and surface hoar can grow where a crust is absent. And, perhaps most
importantly here, a crust that remains near the surface (in the upper few centimeters) for too long
may slowly lose mass and cease to be a crust.

Our data provide strong constraints on models of many of the observed processes.

Surface hoar grew especially at night when relative humidity was high, sometimes with fog, and
with deposition occurring on tent ropes or other surfaces as well as on the snow surface (e.g.
Supplemental Fig. S6), clearly demonstrating a source of vapor from above. Surface hoar
typically formed however, when the upper snow surface was colder than layers beneath,
indicating a vapor source from below. Hence, our surface hoars included elements of both
depositional and sublimation hoar crystals as defined by Gallet et al. [2014] based on
observations at Dome C, East Antarctica (with sublimation growth being the dominant process).

The high density of both single-grain and multi-grain crusts, approaching the density of

ice for the glassy single-grained crusts, requires that the density of the crusts was increased over
time, as wind packing has not been observed to approach these high densities. Crusts form during
days when atmospheric humidity is low, however, and thus when mass is not being added from
above. We have not observed bulk melting at the site (with the one possible exception noted
above), nor do the gas measurements of Orsi et al. [2015] indicate bulk melting, so the density
increase must arise from some combination of vapor diffusion from below and surface or volume
mass transfer likely involving pseudo-liquid layers [Dash et al., 2006], as discussed next.

The data here show that frequently the upper surface is colder than snow beneath, which

will lead to upward mass flux. We lack subcentimetric resolution in thermometry, but physical
understanding indicates that very strong gradients likely develop on the centimeter scale just
below the upper surface during rapid nighttime cooling. Physical understanding, the data here,
and data from previously published studies indicate that intense sunshine generates a temperature
maximum in the snow just below the surface (order of 1 cm) especially in low-density, low-
thermal-conductivity depth hoar [e.g., Alley et al., 1990; Brandt and Warren, 1993], also
contributing to upward vapor transport. Hence, the upper surface is expected to gain mass from
below during the crust- and hoar-forming events [Alley et al., 1990]. Windy conditions would
drive undersaturated air into and out of pore spaces, removing mass, but crusts form during
relatively still times.  The temperature gradients (and noted inversions) measured here at WAIS
Divide (see also Figs. 10 and 11, and Supplemental Fig. S4) are similar to those observed at
GISP2 by Alley et al. [1990] and more than sufficient to move the necessary vapor for crust
development.

We hypothesize here that these surface conditions cause mass fluxes that fill in open

pores in wind-packed layers at the surface to form glazed crusts. A physical model might be
based on the following considerations. The thermal conductivity of ice greatly exceeds that of air,
so heat transport in firn is primarily conductive. Ordinarily, the grain curvature adjacent to pores
tends to cause diffusive mass loss, enlarging pores by filling necks between grains or other
regions of lower vapor pressure. However, because heat flow is primarily through the grain
structure, pores in a surface crust will tend to be colder than interconnected grains when the upper
surface is colder than the firn beneath, favoring mass transport to the pore surfaces, as shown in
Figure 17 [e.g., Sommerfeld, 1983; Fukuzawa and Akitaya, 1993]. Transport may occur by vapor,
surface, or volume diffusion; following Alley and Fitzpatrick [1999], vapor diffusion and surface
transport in premelted films are likely to dominate. Also, mass loss from relatively warm grain
bonds just beneath a growing surface crust by diffusion to the colder crust will tend to lower the
crust, increasing the likelihood that a pore in the crust will move downward to intersect a pre-
existing grain beneath, increasing the crust density.

Due to the intrinsic limitations with the available sensor equipment, and with the sparsity

of usable data for our specific periods of interest, a complete and detailed analysis of radiative
forcings was not completed here. However, to further test our hypothesis and to assess the
accuracy of our measurements, we did execute a simple surface energy budget (SEB) calculation
in order to solve for the ground heat flux term $Q_G$, and ultimately determine if the AWS sensor
data yield flux rates capable of the hypothesized vertical vapor transport in the near-surface snow.
Because data from the AWS-mounted net radiometer and thermistor strings were only available
for the 2012-13 field season, only that specific time window was used for this simple SEB
calculation (see also Figs. 8 and 10).

The surface energy budget represents a balance of turbulent, radiative, and ground heat

fluxes, which are all coupled through various processes [see e.g. Hulth et al, 2010; Miller et al.,
2017]. Because there is no known or observed melting at the WAIS Divide site, and therefore no
phase changes in the near-surface snow, a change in any of the SEB terms is thus balanced by
changes in other terms. For simplicity, we represent this relationship here as:
$$Q_N + Q_S + Q_L + Q_G = 0 \qquad (1)$$

$$Q_N = S_{NET} + L_{NET} = S{\downarrow} + S{\uparrow} + L{\downarrow} + L{\uparrow} \qquad (2)$$

where $Q_N$ is the total net radiation ($S_{NET}$ and $L_{NET}$ are the net short and longwave radiation terms),
$Q_S$ and $Q_L$ are the sensible and latent turbulent heat fluxes respectively, and $Q_G$ is the ground heat
flux. The net radiation term $Q_N$ was determined by combining the short and longwave radiation
data obtained directly from the radiometer (see Fig. 8). Due to limitiations with the Kipp-Zonen
CNR2 sensor, only the radiative *NET* terms were available, and not the individual incoming ($\downarrow$)
and outgoing ($\uparrow$) terms.
Based upon the Monin-Obukhov similarity theory, the sensible and latent heat flux terms
can be expressed as:
$$Q_S = \rho c_p u_* T_* \ and \ Q_L = \rho L_S u_* q_* \tag{3}$$
where $\rho$ denotes air density, $c_p$ is the specific heat of dry air at constant pressure (1005 J K$^{-1}$ kg$^{-1}$),
and $L_S = 2.83 \cdot 10^6$ J kg$^{-1}$ is the latent heat of sublimation. We use bulk method approximations for
the turbulent scales of wind speed ($u_*$), temperature ($T_*$), and humidity ($q_*$), and their related
stability correction functions [Van As et al., 2005; Andreas, 2002; Fairall et al., 1996; Holtslag
and DeBruin,1988]. We also employed an optimal velocity roughness length (~0.03 mm) and
calculated the related roughness terms using published equations [Miller et al., 2017; Van As et
al., 2005; Andreas, 1987]. As previously noted, relative humidity values reported here are
expressed in terms of saturation vapor pressure over ice and corrected for low-temperature offsets
[see Anderson, 1994]. Specific humidity is calculated from relative humidity using published
equations [Van As et al., 2005].
Results of this SEB calculation are shown in Figure 18, and values for ground heat flux
were were determined by solving equation (1) for $Q_G$. Over the ~24 hr, low-wind period shown
highlighted in Figure 8 that features a surface glaze (labeled 'b'), the net ground heat flux $Q_G$
does corroborate a condition favorable for upward (negative) energy flux, particularly during the
morning hours of the 24-DEC-2012.
Using our measured snow pit density data (see Fig. 14), combined with published
calcuations for conductivity, we calculated an average value of $0.35 \pm 0.05$ Wm$^{-1}$ K$^{-1}$ for the
thermal conductivity ($K$) of the upper most layers of snow, assuming a linear snow compaction
factor [Miller et al., 2017; Jordan, 1991]. Then, using our calculated ground heat flux data above
($Q_G$) combined with the equation for the sub-surface heat flux,
$$Q_G = -K \frac{\Delta T}{\Delta z} \tag{4}$$
we calculated empirical estimates for the vertical temperature gradient ($\Delta T$), over a 20 cm depth
($\Delta Z$) interval [Van As et al., 2005].

Our results yield an average vertical temperature gradient of $\sim 3.6 \pm 0.6$ ºC over the ~24

hr, low-wind period shown highlighted in Figure 8 which features the surface glaze. This result is
consistent with the direct thermistor string data (see Figs. 10 and 11) which indicate a gradient
between the near-surface air sensor (AWS), and the shallowest in-firn snow sensor (S1) of ~3.0ºC
during the peak of the inversion and glazing episode on 24-DEC-2012.

We realize that this energy balance investigation is simplistic and makes several

assumptions, however we do believe that our calculated ground heat flux rates are sufficient to
drive the necessary vapor mass transport needed for glazed-crust development [Pinzer et al.,
2012]. While it is outside the scope of this study, a broader and more thorough investigation into
the overall radiative and SEB responses, boundary-layer stability responses, cloud forcings, and
vapor mass flux rates, is warranted in order to better quantify and validate these results.

Although summertime crusts dominate in the ice core, many wintertime crusts were

identified, raising additional questions. We lack direct observations in winter, and so can only
speculate on mechanisms active then. However, the basic picture drawn above for summertime
crusts may also apply in winter. The lower temperatures, and lack of intense solar heating, make
crust formation less likely. However, stronger wintertime winds would allow greater wind-
packing of the upper surface, producing fewer and smaller pores to be filled to make a thin crust,
and thus making crust formation easier. Although accumulation is more-or-less evenly distributed
through the year, we speculate (based upon variability observed in AWS data) that there may be
extended intervals up to weeks in length during the winter when the surface is relatively stable,
partially or completely offsetting the slower mass transport from colder temperatures.
Furthermore, the AWS data show that mid-winter temperatures have risen as high as -15°C
during strong warming events accompanied by high winds ($> 10$ m s$^{-1}$), and likely linked to
transport of air masses from the coast. Such warm air masses paired with these high winds, would
produce relatively high vapor pressures, contribute to greater surface packing, and promote
temperature inversions and upward near-surface vapor flux during the subsequent cooling.

The great abundance of crusts at WAIS Divide compared to other ice cores we have

studied may be because conditions are "just right" at WAIS Divide. We have observed loss of a
wind-packed crust at WAIS Divide, and also at GISP2 in central Greenland; the strong mass loss
from ~1 cm down in the snowpack is not conducive to long-term survival of any crust there [e.g.,
Alley et al., 1990]. Low but nonzero summertime accumulation thus may lead to loss of crusts,
whereas higher accumulation after formation buries them below that zone of mass loss and so
allows their preservation. The large wintertime variability and high wintertime temperatures at
WAIS Divide may be important in generating sufficiently high mass fluxes to produce wintertime
crusts.

At least in summertime, crusts do seem to record a particular meteorological pattern of

storms alternating with still conditions. The time-series of frequency of occurrence of crusts thus
would be affected by a change in the frequency of occurrence of these conditions. Turning this
into a paleoclimatic indicator would require additional steps, however, as the frequency of
preserved crusts could decrease because fewer were formed or because more were destroyed,
with different causes. Information on changing frequency of meteorological events might be
useful [e.g., Hammer, 1985; Alley, 1988]. We believe that the clear association of crust formation
with particular events, and the clear trends in crust occurrence in the core, motivate additional
research on topics including crust formation in non-summer seasons, but we do not know whether
this ultimately could yield a valuable paleoclimatic indicator.

**5: Conclusions**


Summertime observations at the WAIS Divide site show that prominent visible strata

form at or very near the surface during summer, by processes that typically are repeated a few
times during each summer. A storm produces a wind-packed layer. The following high-pressure
system brings light winds, warm days and cool nights, strong sunshine, and low relative
humidity. High-density, single-grain-thick glazed crusts preferentially form at the surface during
these high-pressure intervals, in as little as a single day, and then strengthen and evolve. Crusts
are extensive, although typically broken by sub-meter or few-meter uncrusted regions spaced tens
of meters to more than 100 m apart. Daytime solar heating drives upward mass transport to crusts
from developing depth hoar beneath, strengthening the crusts. A simple surface energy budget
(SEB) calculation shows that sufficient vertical heat fluxes exist to explain both the observed
near-surface temperature inversions, and the vapor mass-flux necessary for the associated glazed-
crust formation. After formation, crusts are broken by polygonal cracks extending typically 20-30
cm deep, likely from contraction during nighttime cooling. Relative humidity then rises in the air
above, contributing to growth of surface hoar during nighttime cooling. Subsequent storms
typically bury the crust-hoar complexes, although crusts can be lost during evolving surface
conditions if not buried below the top one to a few centimeters.

Study of the WAIS Divide deep core shows that crusts are preserved through the bubbly

ice. Crusts are most common in layers deposited during summertime, but also occur in winter
accumulation. Study of AWS data suggests that the intrusion of warm coastal air during winter
may generate strong temperature gradients, which may contribute to wintertime crust formation
in wind-packed layers.

The frequency of occurrence of crusts in the core varies with time, suggesting the

possibility that crusts could be used as a paleoclimatic indicator.  However, additional work
would be required, including addressing whether crust frequency varies because of changes in
formation or changes in destruction of crusts previously formed.  The crusts do not produce
significant anomalies in other ice-core paleoclimatic records, likely at least in part because they
are discontinuous and broken by contraction cracks.
**6: Data Availability:**
Data policy: All data presented here are available via download from NSIDC
(http://nsidc.org) or from the WAIS Divide data portal (http://waisdivide.unh.edu).

**7: Author Contribution:**
A.J. Orsi assisted with field observations and experiments. A. Muto designed the near-
surface PRD sensor strings and developed the associated logging code. M. Spencer documented
all ice-core crust observations during the WAIS Divide core processing at the National Ice Core
Laboratory. J.M. Fegyveresi and R.B. Alley prepared the manuscript with contributions from all
co-authors.

**8: Acknowledgements:**

We acknowledge the following funding sources for support of this work: U.S. National

Science Foundation Division of Polar Programs grants 0539578, 1043528, 1142085, 1619793.
We also acknowledge Donald E. Voigt, Joan J. Fitzpatrick, Eric D. Cravens, and the staff of the
U.S. National Ice Core Laboratory in Denver, Colorado, as well as the WAIS Divide Science
Coordination Office at the University of New Hampshire, and the Ice Drilling Design and
Operations group at the University of Wisconsin. We thank numerous colleagues involved with
the WAIS Divide project, especially Kendrick Taylor, Mark Twickler, and Joseph Souney. We
thank Bess Koffman, Gifford Wong, Dominic Winski, Aron Buffen, and Logan Mitchell for
assistance with snow pit preparation. We thank Jonathan Thom and the University of Wisconsin-
Madison Automatic Weather Station Program for assistance with weather station sensor
installation. Lastly, we thank our reviewers, whose thoughtful suggestions and questions served
to clarify and improve this manuscript. Any use of trade, firm, or product names is for descriptive
purposes only and does not imply endorsement.


**9: References**


Alley, R.B., 1988. Concerning the deposition and diagenesis of strata in polar firn. *Journal of*
*Glaciology*, 34: 283-290.
Alley, R.B., Saltzman, E.S., Cuffey, K.M. and Fitzpatrick, J.J., 1990. Summertime formation of
depth hoar in central Greenland. *Geophysical Research Letters*, *17*(13): 2393-2396,
doi:10.1029/GL017i013p02393.
Alley, R.B. and Fitzpatrick, J.J., 1999. Conditions for bubble elongation in cold ice-sheet ice.
*Journal of Glaciology*, *45*(149): 147-153.
Alley, R.B., Shuman, C.A., Meese, D.A., Gow, A.J., Taylor, K.C., Cuffey, K.M., Fitzpatrick, J.J.,
Grootes, P.M., Zielinski, G.A., Ram, M. and Spinelli, G., 1997. Visual–stratigraphic
dating of the GISP2 ice core: Basis, reproducibility, and application. *Journal of*
*Geophysical Research: Oceans*, *102*(C12): 26367-26381, doi:10.1029/96JC03837.
Anderson, P.S., 1994. A method for rescaling humidity sensors at temperatures well below
freezing. *Journal of Atmospheric and Oceanic Technology*, *11*(5), 1388-1391.
Anderson, D. L., and Benson C.S., 1963, The densification and diagenesis of snow, in Ice and
Snow: Properties, Processes and Applications, edited by W. D. Kingery, pp. 391–411,
MIT Press.
Andreas, E.L., 1987. A theory for the scalar roughness and the scalar transfer coefficients over
snow and sea ice. Boundary-Layer Meteorology, 38(1), 159-184.
Andreas, E.L., 2002. Parameterizing scalar transfer over snow and ice: a review. Journal of
Hydrometeorology, 3(4), 417-432.
Banta, J.R., McConnell, J.R., Frey, M.M., Bales, R.C. and Taylor, K., 2008. Spatial and temporal
variability in snow accumulation at the West Antarctic Ice Sheet Divide over recent
centuries. *Journal of Geophysical Research: Atmospheres*, *113*(D23), doi:
10.1029/2008JD010235.
Battle, M.O., Severinghaus, J.P., Sofen, E.D., Plotkin, D., Orsi, A.J., Aydin, M., Montzka, S.A.,
Sowers, T. and Tans, P.P., 2011. Controls on the movement and composition of firn air at
the West Antarctic Ice Sheet Divide. *Atmospheric Chemistry and Physics*, *11*(21), 11007-
11021, doi:10.5194/acp-11-11007-2011.
Benson, C.S., 1962. Stratigraphic Studies on Greenland Ice Sheet and a Quantitative
Classification of Glaciers. *Bulletin of the American Meteorological Society*, *43*(4): 141.
Brandt, R.E. and Warren, S.G., 1993. Solar-heating rates and temperature profiles in Antarctic
snow and ice. *Journal of Glaciology*, *39*(131), 99-110.
Buizert, C. et al., 2015. The WAIS Divide deep ice core WD2014 chronology–Part 1: Methane
synchronization (68–31 ka BP) and the gas age–ice age difference. *Climate of the Past*,
*11*(2): 153-173, doi:10.5194/cp-11-153-2015.
Burgener, L., Rupper, S., Koenig, L., Forster, R., Christensen, W.F., Williams, J., Koutnik, M.,
Miege, C., Steig, E.J., Tingey, D., Keeler, D., and Riley, L., 2013. An observed negative
trend in Antarctic accumulation rates from 1975 to 2010: Evidence from new observed
and simulated records. *Journal of Geophysical Research-Atmospheres*, 118(10): 4205-
4216.
Champollion, N., Picard, G., Arnaud, L., Lefebvre, E. and Fily, M., 2013. Hoar crystal
development and disappearance at Dome C, Antarctica: observation by near-infrared
photography and passive microwave satellite. *The Cryosphere*, *7*(4): 1247-1262,
doi:10.5194/tc-7-1247-2013.
Colbeck, S.C., 1982. An overview of seasonal snow metamorphism. *Reviews of Geophysics*,
*20*(1): 45-61.

Colbeck, S.C., 1983. Theory of metamorphism of dry snow. *Journal of Geophysical Research:*
*Oceans*, *88*(C9): 5475-5482.
Conger, S.M., McClung, D.M., 2009. Comparison of density cutters for snow profile
observations. *Journal of Glaciology*, 55(189): 163-169.
Criscitiello, A.S., Das, S.B., Karnauskas, K.B., Evans, M.J., Frey, K.E., Joughin, I., Steig, E.J.,
McConnell, J.R., and Medley, B., 2014. Tropical Pacific Influence on the Source and
Transport of Marine Aerosols to West Antarctica. *Journal of Climate*, 27(3): 1343-1363.
Das, S.B. and Alley, R.B., 2005. Characterization and formation of melt layers in polar snow:
observations and experiments from West Antarctica. *Journal of Glaciology*, *51*(173):
307-312, doi:10.3189/172756505781829395.
Dash, J.G., Rempel A.W., and Wettlaufer J.S., 2006. The physics of premelted ice and its
geophysical consequences, *Rev. Mod. Phys*. 78, 695-741,
doi:10.1103/RevModPhys.78.695.
Fairall, C.W., Bradley, E.F., Rogers, D.P., Edson, J.B. and Young, G.S., 1996. Bulk
parameterization of air-sea fluxes for tropical ocean-global atmosphere coupled-ocean
atmosphere response experiment. Journal of Geophysical Research: Oceans, 101(C2),
3747-3764.
Fegyveresi, J.M., 2015, Physical properties of theWest Antarctic Ice Sheet (WAIS) Divide deep
core: Development, evolution, and interpretation, PhD thesis, The Pennsylvania State
Univ., State College, Pa.
Fegyveresi, J.M., Alley, R.B., Fitzpatrick, J.J., Cuffey, K.M., McConnell, J.R., Voigt, D.E.,
Spencer, M.K. and Stevens, N.T., 2016. Five millennia of surface temperatures and ice
core bubble characteristics from the WAIS Divide deep core, West Antarctica.
*Paleoceanography*, *31*: 416–433, doi:10.1002/2015PA002851.
Fitzpatrick, J.J., Voigt, D.E., Fegyveresi, J.M., Stevens, N.T., Spencer, M.K., Cole-Dai, J., Alley,
R.B., Jardine, G.E., Cravens, E.D., Wilen, L.A. and Fudge, T.J., 2014. Physical
properties of the WAIS Divide ice core. *Journal of Glaciology*, *60*(224), 1181-1198,
doi:10.3189/2014JoG14J100.
Fukuzawa, T. and Akitaya E., 1993. Depth-Hoar Crystal-Growth in the Surface-Layer under
High-Temperature Gradient. *Annals of Glaciology*, *18*: 39-45.
Gallet, J.C., Domine, F., Savarino, J., Dumont, M. and Brun, E., 2014. The growth of sublimation
crystals and surface hoar on the Antarctic plateau. *The Cryosphere*, *8*(4): 1205-1215,
doi:10.5194/tc-8-1205-2014.
Gow, A. 1965. On the accumulation and seasonal stratification of snow at the South Pole. *Journal*
*of Glaciology*, *5*(40): 467-477.
Gow, A. 1969. On the rates of growth of grains and crystals in south polar firn. *Journal of*
*Glaciology*, *8*(53): 241-252.
Hammer, C.U., 1985. The influence on atmospheric composition of volcanic eruptions as derived
from ice-core analysis. *Annals of Glaciology*, *7*: 125-129.
Holtslag, A.A.M. and De Bruin, H.A.R., 1988. Applied modeling of the nighttime surface energy
balance over land. Journal of Applied Meteorology, 27(6), 689-704.
Hulth, J., Rolstad, C., Trondsen, K. and Rødby, R.W., 2010. Surface mass and energy balance of
Sørbreen, Jan Mayen, 2008. Annals of Glaciology, 51(55), 110-119.
Jordan, R.: A one-dimensional temperature model for a snow cover: Technical documentation for
SNTHERM.89, Special Report 91-16, Cold Regions Research and Engineering
Laboratory (U.S.) and Engineer Research and Development Center (U.S.), 1991.
Koerner, R.M. 1971. A stratigraphic method of determining the snow accumulation at Plateau
Station, Antarctica, and application to South Pole-Queen Maud Land traverse 2, 1965-

1966. In Crary, A.P. *Antarctic snow and ice studies II*, (Washington, DC, American
Geophysical Union): 225-238.
Koffman, B.G., Kreutz, K.J., Breton, D.J., Kane, E.J., Winski, D.A., Birkel, S.D., and Kurbatov,
595          A.V., 2014. Centennial-scale variability of the Southern Hemisphere westerly wind belt
in the eastern Pacific over the past two millennia. *Climate of the Past*, 10(3): 1125-1144.
Lazzara, M.A., Weidner, G.A., Keller, L.M., Thom J.E., and Cassano, J.J., 2012. Antarctic
Automatic Weather Station Program 30 Years of Polar Observations. *Bulletin of the*
*American Meteorological Society*, *93*(10): 1519-1537, doi:10.1175/BAMS-D-11-
00015.1.
Miller, N.B., Shupe, M.D., Cox, C.J., Noone, D., Persson, P.O.G. and Konrad, S., 2017. Surface
energy budget responses to radiative forcing at Summit, Greenland. The Cryosphere,
11(1), 497-516.
Mitchell, L.E., Buizert, C., Brook, E.J., Breton, D.J., Fegyveresi, J., Baggenstos, D., Orsi, A.,
Severinghaus, J., Alley, R.B., Albert, M. and Rhodes, R.H., 2015. Observing and
modeling the influence of layering on bubble trapping in polar firn. *Journal of*
*Geophysical Research: Atmospheres*, *120*(6): 2558-2574, doi:10.1002/2014JD022766.
Muto, A., Scambos, T.A., Steffen, K., Slater, A.G. and Clow, G.D., 2011. Recent surface
temperature trends in the interior of East Antarctica from borehole firn temperature
measurements and geophysical inverse methods. *Geophysical Research Letters*, *38*(15),
doi:10.1029/2011GL048086.
Orsi, A.J., Kawamura, K., Fegyveresi, J.M., Headly, M.A., Alley, R.B. and Severinghaus, J.P.,
2015. Differentiating bubble-free layers from melt layers in ice cores using noble gases.
*Journal of Glaciology*, *61*(227): 585-594, doi:10.3189/2015JoG14J237.
Pinzer, B.R., Schneebeli, M. and Kaempfer, T.U., 2012. Vapor flux and recrystallization during
dry snow metamorphism under a steady temperature gradient as observed by time-lapse
micro-tomography. The Cryosphere, 6(5), 1141-1155.
Sigl, M. et al., 2016. The WAIS Divide deep ice core WD2014 chronology–Part 2: Annual-layer
counting (0–31 ka BP). *Climate of the Past*, *12*(3): 769-786, doi:10.5194/cp-12-769-
2016.
Sommerfeld, R.A., 1983. A branch grain theory of temperature gradient metamorphism in snow.
*Journal of Geophysical Research: Oceans*, *88*(C2): 1484-1494,
doi:10.1029/JC088iC02p01484.
Sorge, E., 1935. Glaziologische Untersuchungen in Eismitte. *Brockamp, B., and others.*
*Glaziologie. Leipzig, FA Brockhaus*, *935*: 62-270.
Van As, D., Van Den Broeke, M. and Van De Wal, R., 2005. Daily cycle of the surface layer and
energy balance on the high Antarctic Plateau. Antarctic Science, 17(1), 121-133.
WAIS Divide Project Members, 2013. Onset of deglacial warming in West Antarctica driven by
local orbital forcing. *Nature*, *500*(7463): 440-444, doi:10.1038/nature12376.
Weller, G. 1969. The heat and mass balance of snow dunes on the central Antarctic Plateau.
*Journal of Glaciology*, *8*: 277-284.

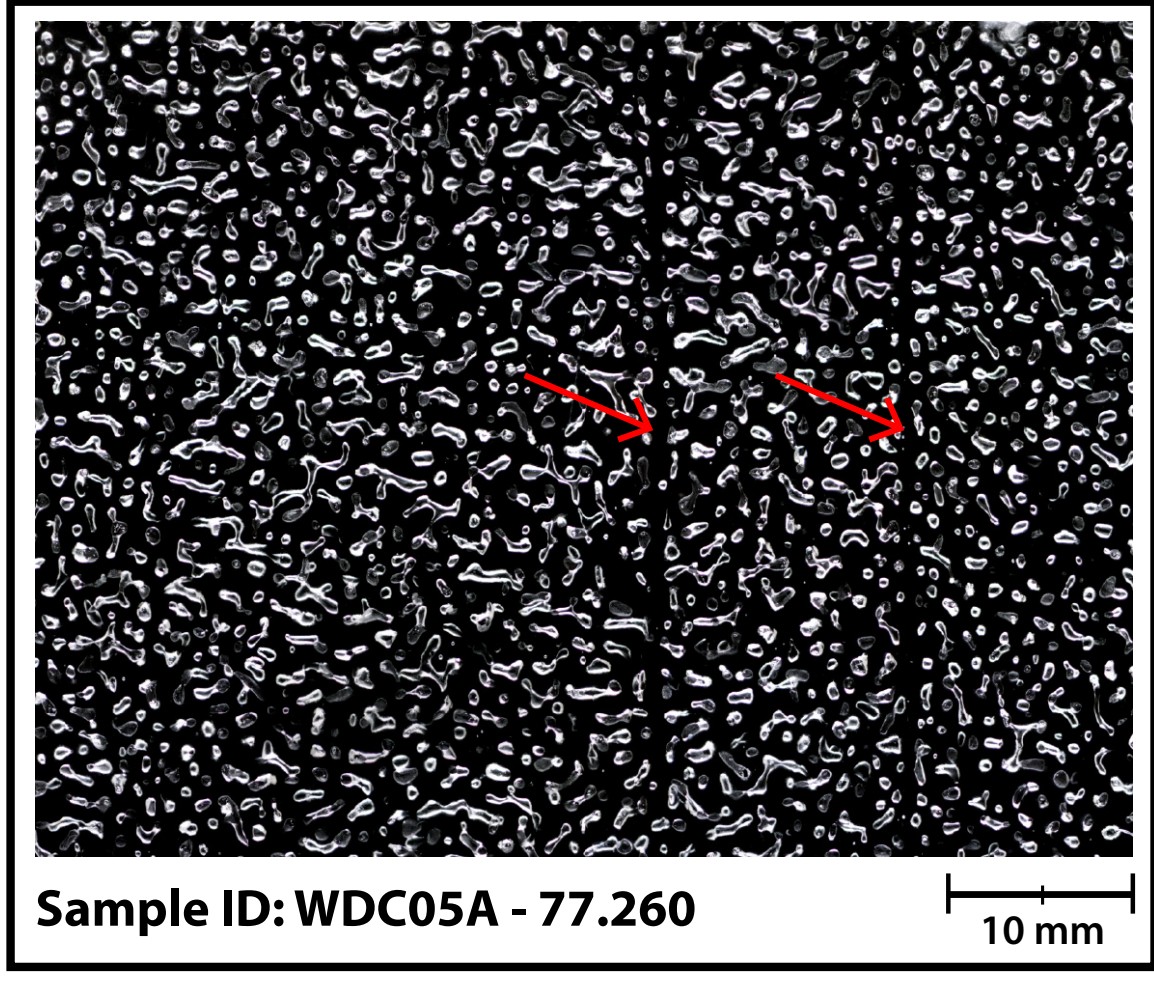

Sample ID: WDC05A - 77.260

10 mm

**Figure 1:** A thick-section image of a sample prepared from a depth of ~77.260 meters showing
two preserved crusts. Both layers are ~1 mm thick and appear mostly bubble-free. All bubbles
here appear white, with the surrounding ice black. The general elongated shape of the bubbles is
due the proximity of this sample to the bubble close-off depth at WAIS Divide of ~75 meters).
This sample is from the secondary WDC05A core at the WAIS Divide site. Image modified from
Orsi et al. [2015].

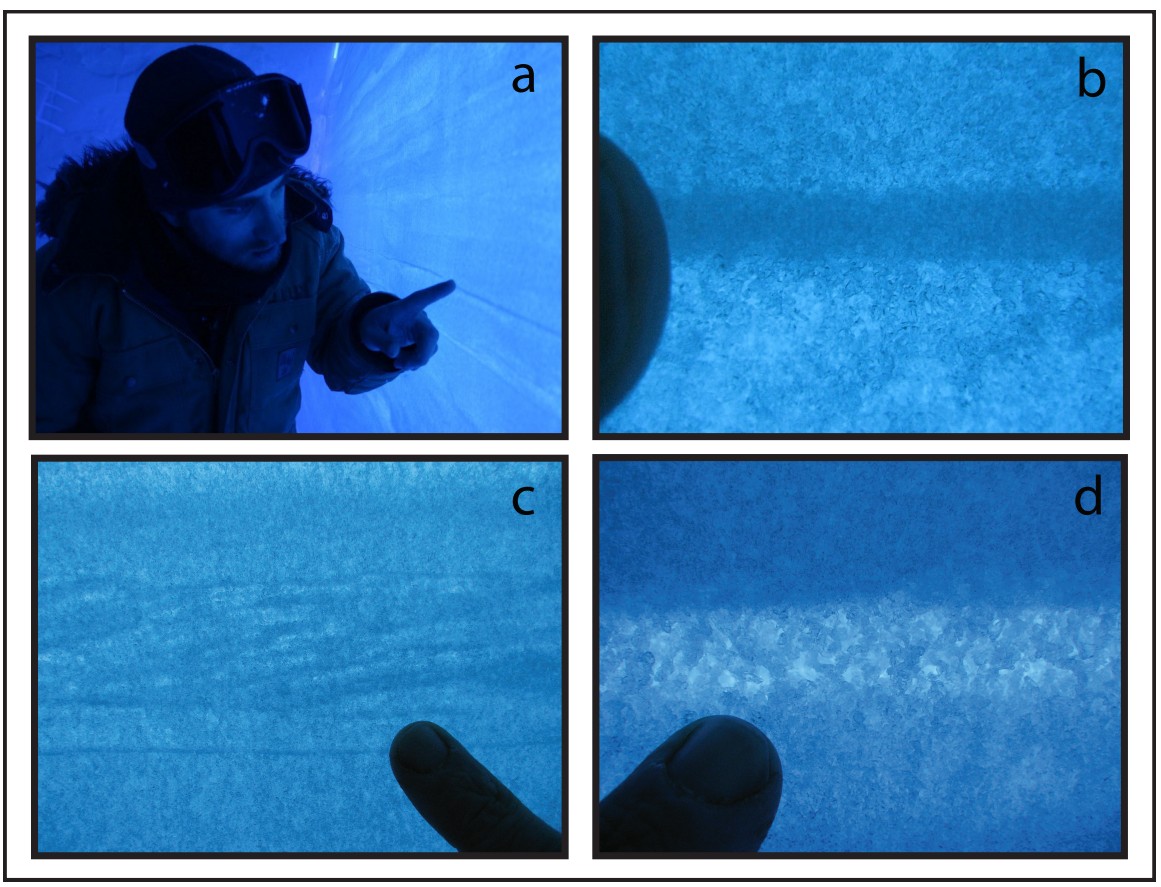

**Figure 2:** The lead author in a 2-meter snow pit prepared at WAIS Divide (pit 2009-10-A).
Multi-grain crusts (a, b), preserved sastrugi with cross-bedding (c), and hoar layers (d) are all
easily identifiable.

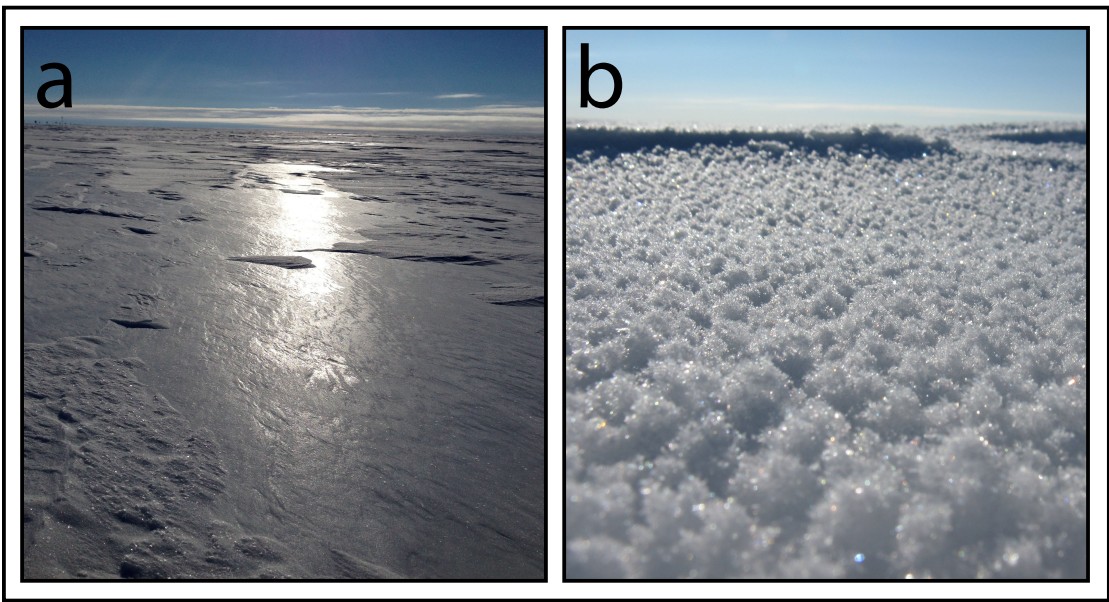

Figure 3: Surface "glaze" (a) that formed on a calm, sunny day (23-Dec-2012) at WAIS Divide, and the subsequent surface hoar layer (b) that formed on its surface after several calm days.

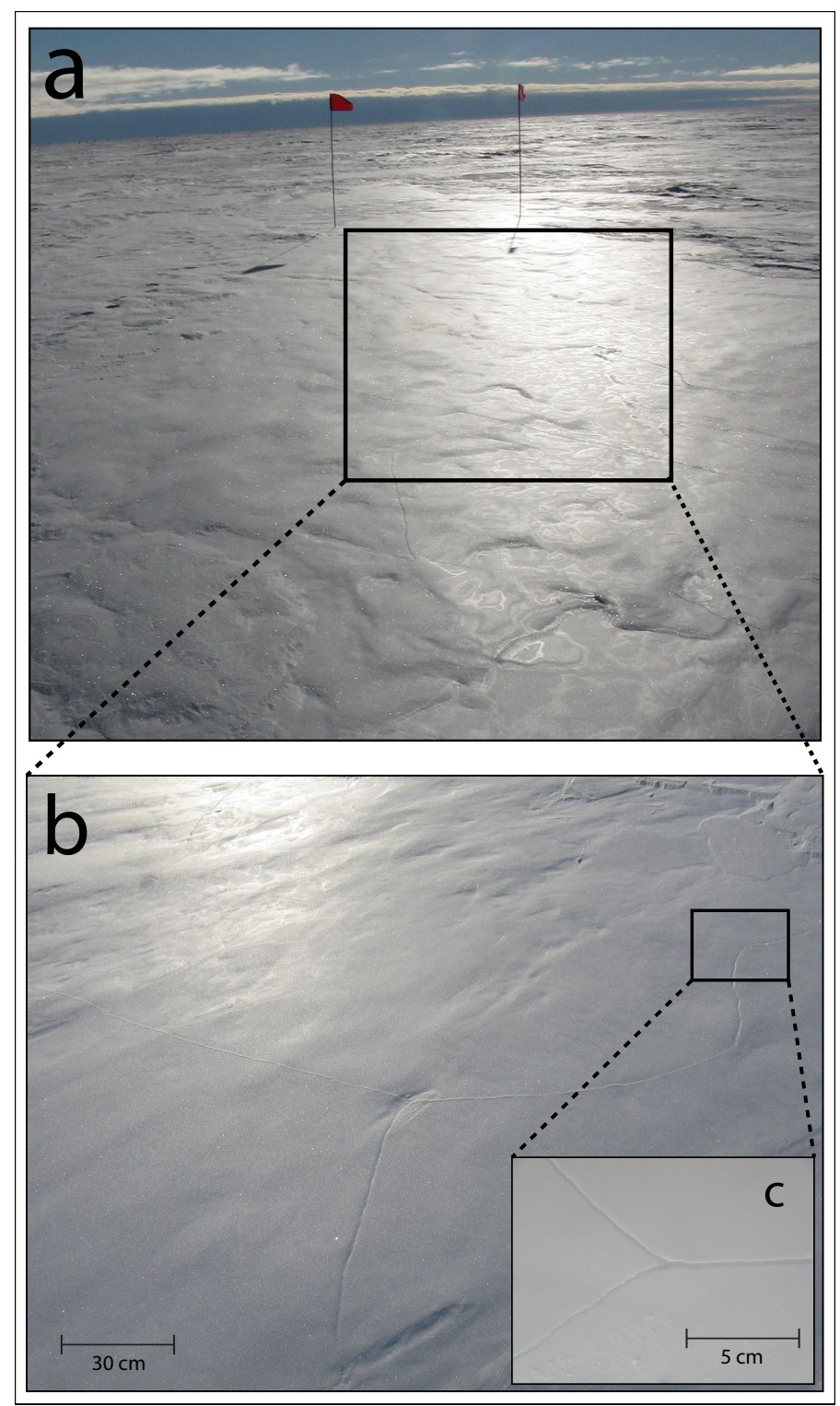

**Figure 4:** Surface "glaze" seen at the WAIS Divide site. (a). A zoomed-in view shows the
polygonal cracking that initiates at the surface from thermal contraction, following several sunny,
clear-sky days (b). Closer inspection reveals greater detail and scale of a crack triple-junction (c).

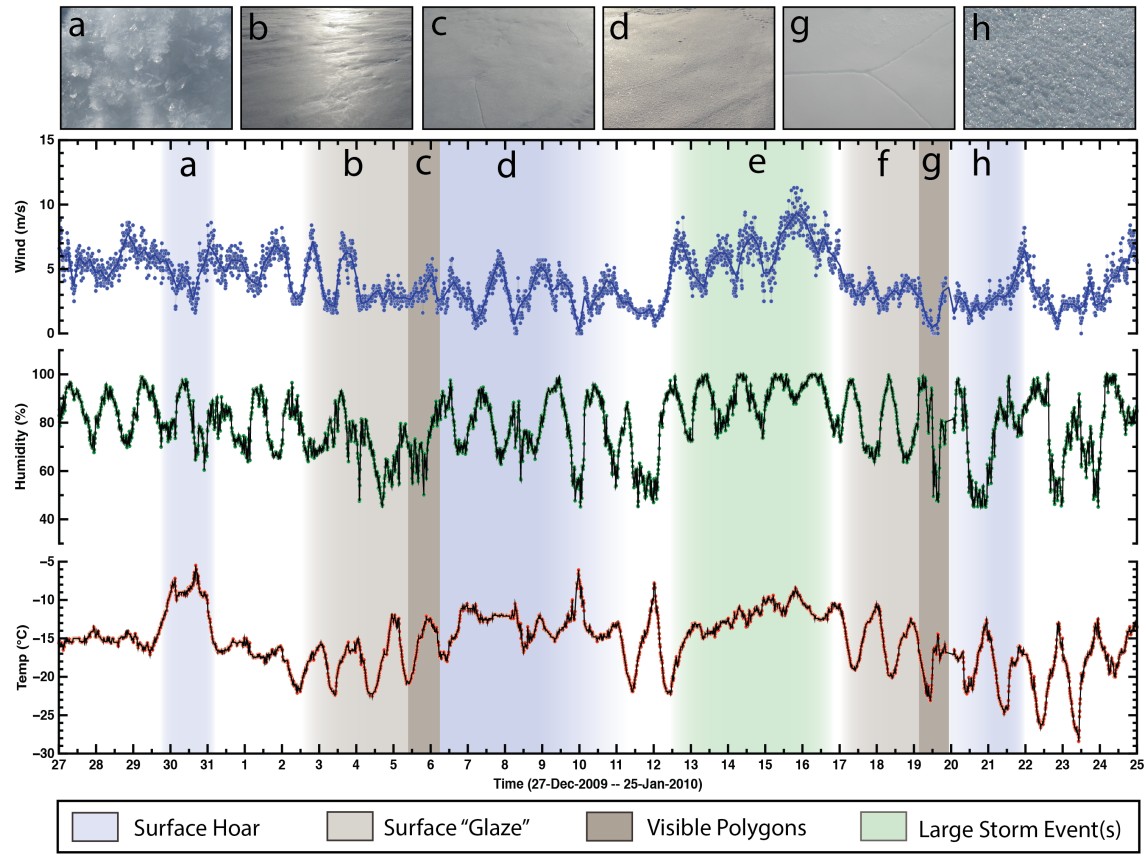

**Figure 5:** Surface evolution over 29 days in 2009-10 season, and AWS data. Shading shows
episodes of surface hoar, glazes, and polygonal cracking; storm events are also shown. Letters
near the top refer to photographs above of specific features or events. All dates and times are
GMT (-12 WAIS local time). The errors for all AWS instruments are listed in Supplemental
Table S1.

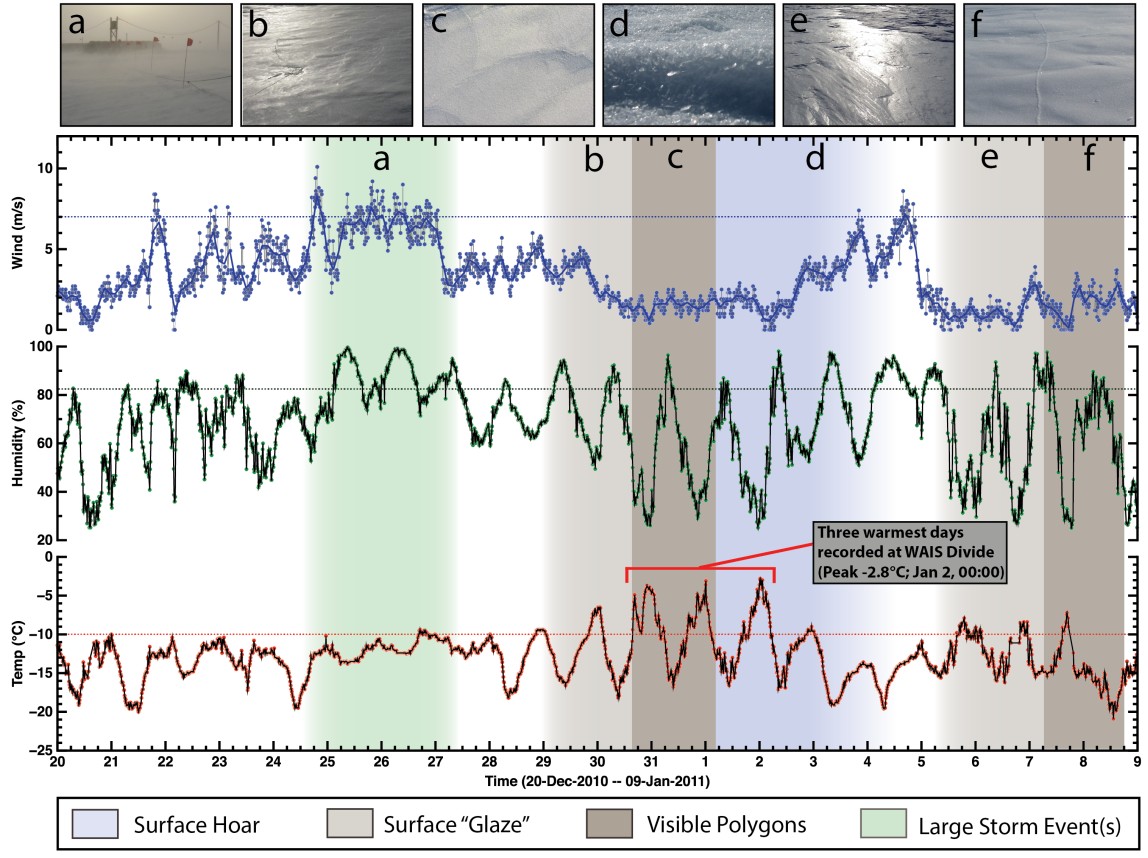

**Figure 6:** Surface evolution over 20 days in 2010-11 season, and AWS data. Shading shows episodes of surface hoar, glazes, and polygonal cracking; storm events are also shown. Letters near the top refer to photographs above of specific features or events. All dates and times are GMT (-12 WAIS local time). The errors for all AWS instruments are listed in Supplemental Table S1.

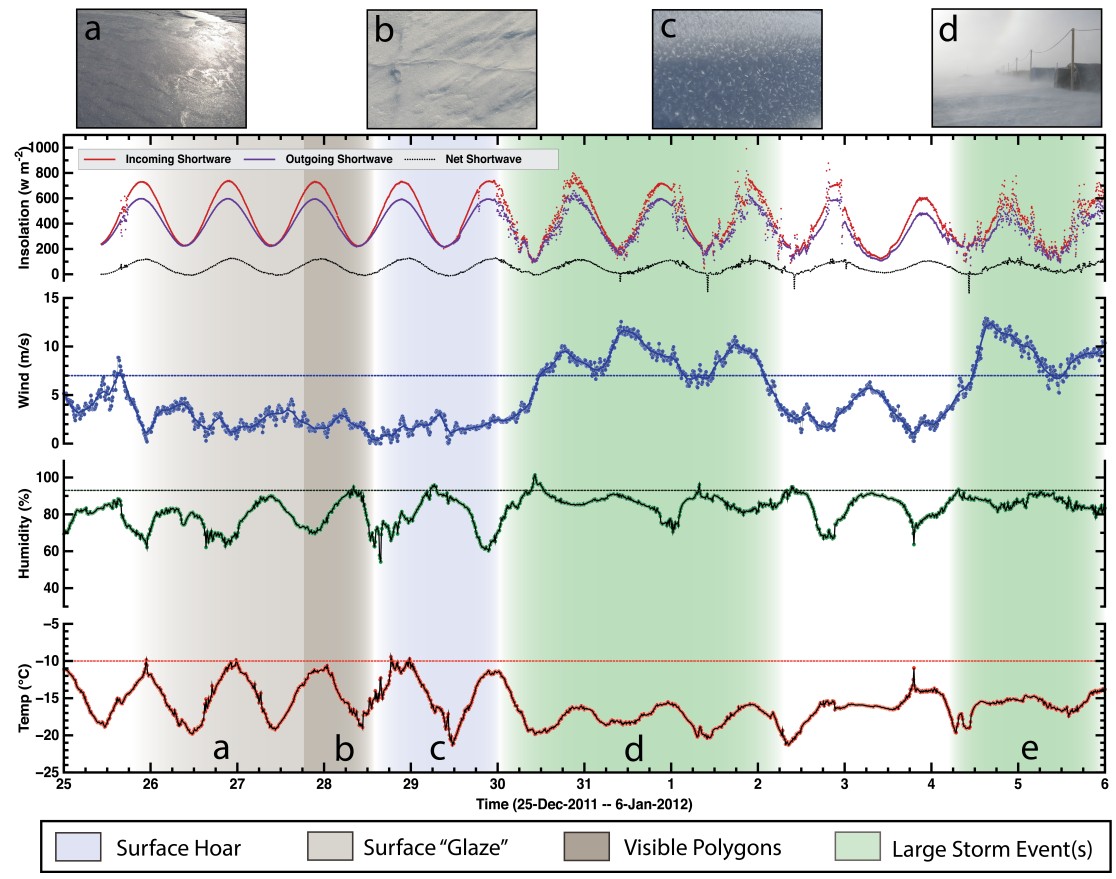

**Figure 7:** Surface evolution over 12 days in 2011-12 season, and AWS data. Shading shows
episodes of surface hoar, glazes, and polygonal cracking; storm events are also shown. Letters
near the top refer to photographs above of specific features or events. All dates and times are
GMT (-12 WAIS local time). The errors for all AWS instruments are listed in Supplemental
Table S1.

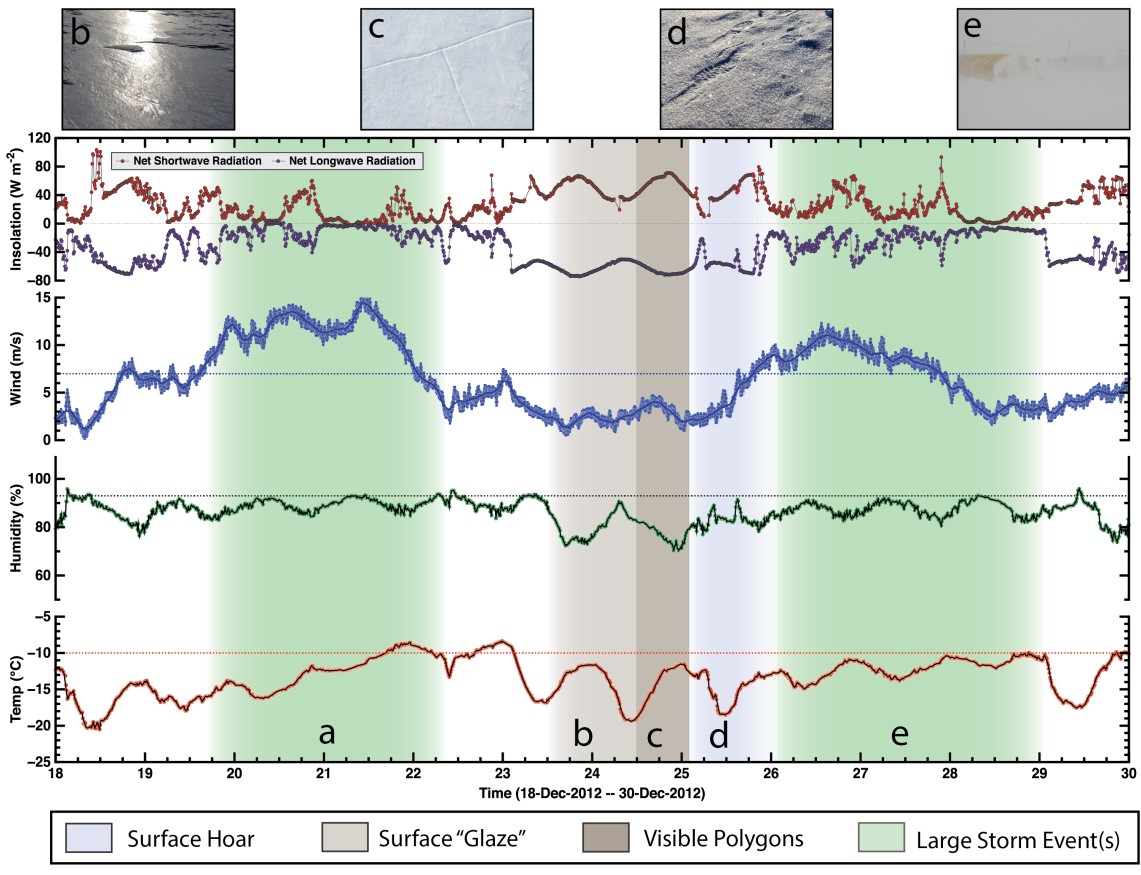

**Figure 8:** Surface evolution over 12 days in 2012-13 season, and AWS data. Shading shows episodes of surface hoar, glazes, and polygonal cracking; storm events are also shown. Letters near the top refer to photographs above of specific features or events. All dates and times are GMT (-12 WAIS local time). The errors for all AWS instruments are listed in Supplemental Table S1.

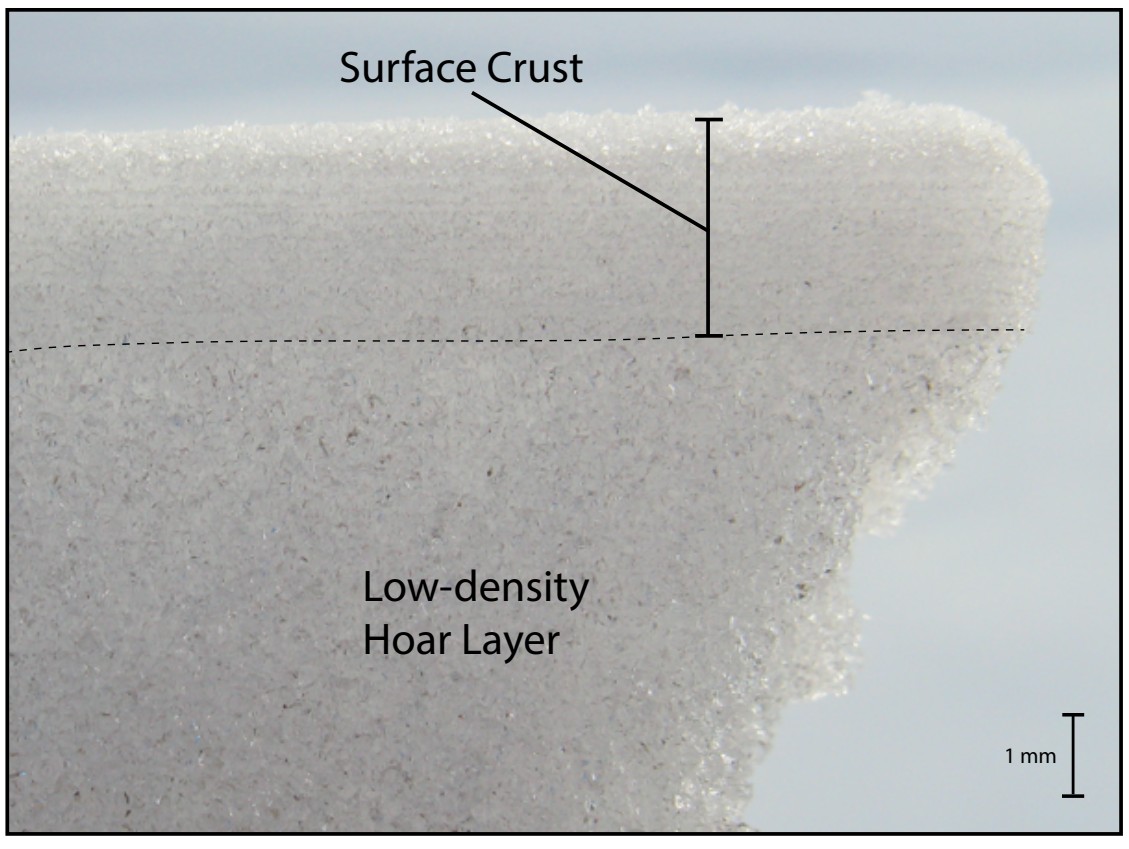

**Figure 9:** A firn sample excavated from a glazed area at WAIS Divide before the onset of
polygonal cracking, showing a couplet of an evolved high-density ($> 400$ kg m$^{-3}$), ~3 mm multi-
grain surface crust containing single-grain crusts, and overlying a lower-density ($< 300$ kg m$^{-3}$)
hoar layer.

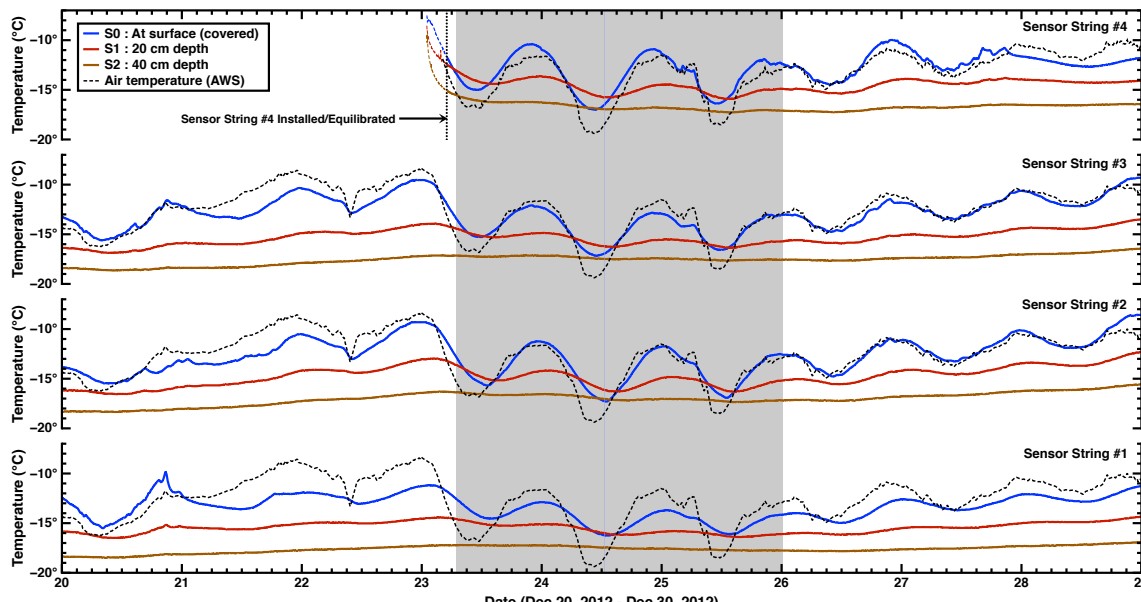

**Figure 10:** Temperature measurements (1 min interval) in firn from the 2012-13 season, from the
upper-most three PRDs (surface down to 40 cm). Data are from the four sensor stations closest to
the station. The shaded area corresponds to an episode of glaze and hoar growth (see Fig. 8).
Distinct near-surface temperature inversions occurred each night during this 3-day period (see
Fig. 11). Sensor #4 was not installed until Dec 22[nd], and therefore did not equilibrate until early
on the 23[rd] as indicated. Air temperature is also shown as recorded by the AWS (errors listed in
Supplemental Table S1). The AWS temperature sensor is located ~1 meter above the snow
surface. All dates and times are GMT (-12 WAIS local time).

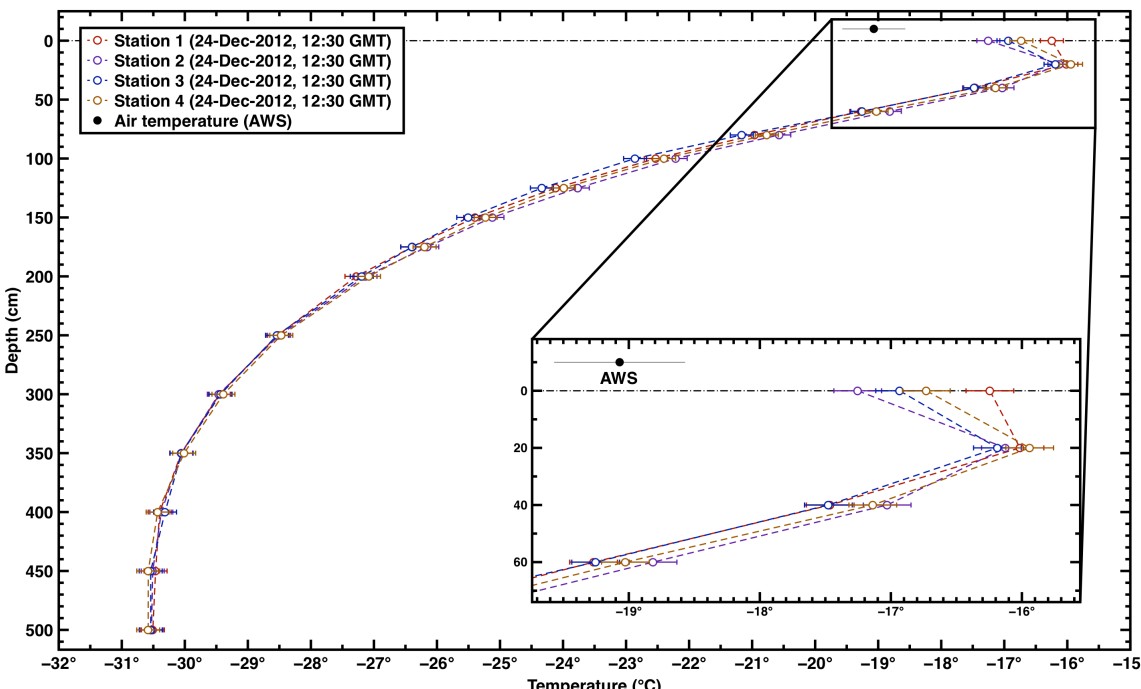

**Figure 11:** Snap-shot temperature readings for PRD-string stations #1-4, taken on 24-Dec-2012
at ~12:30 GMT, showing the temperature inversion with colder air (AWS data) and upper surface
over warmer near-surface snow.

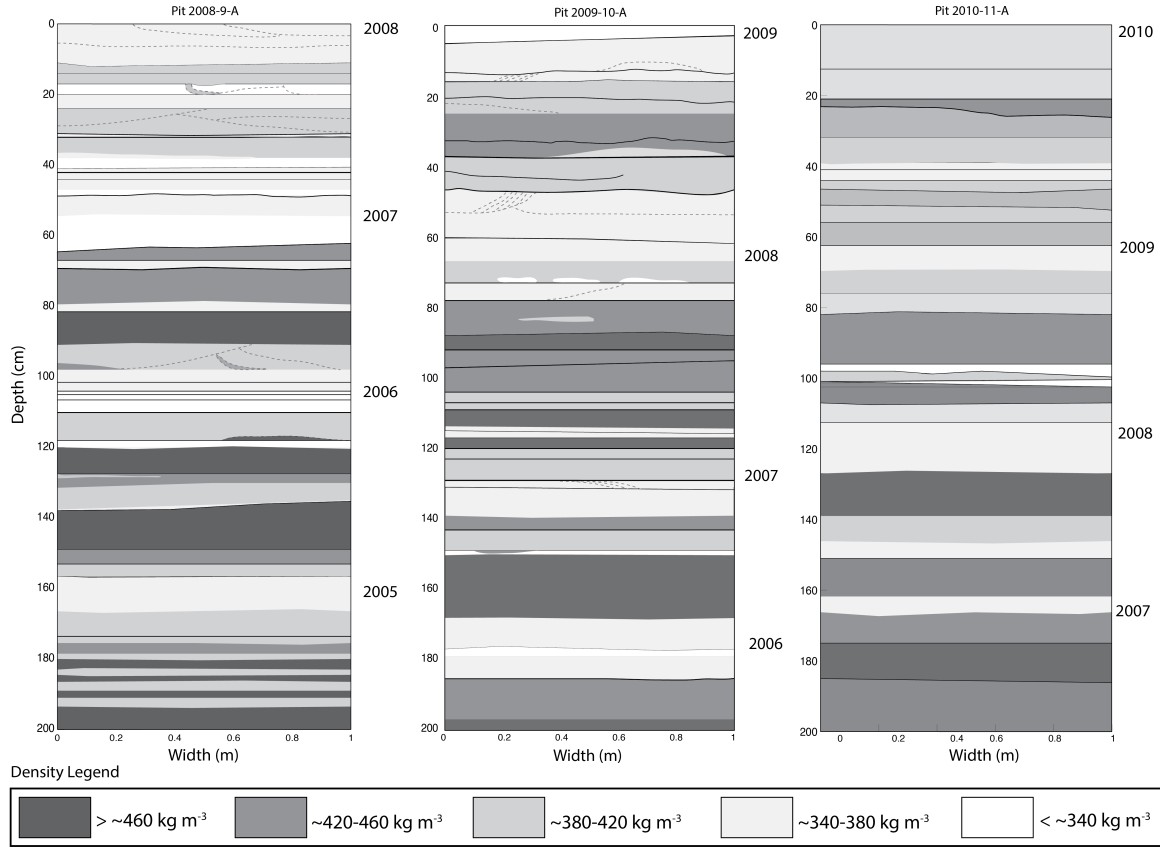

**Figure 12:** Complete wall maps of back-lit snow pits prepared during 2008-09, 2009-10, and
2010-11 WAIS Divide field seasons. Layering and density contrast are noted by degree of
shading. Fine- to medium- grained, higher-density snow/firn layers are shown with darker grey
coloring, whereas coarse-grained and low-density layers (e.g., depth hoar) are shown in white.
Crusts are indicated with solid lines, while dotted lines are used to represent cross-bedding at
depth. Years were identified based on approximate depths of peak summers and the average
measured densities. The pit wall surfaces trend in parallel with the prevailing wind direction at
WAIS Divide (approximately north-south, with north to the right).

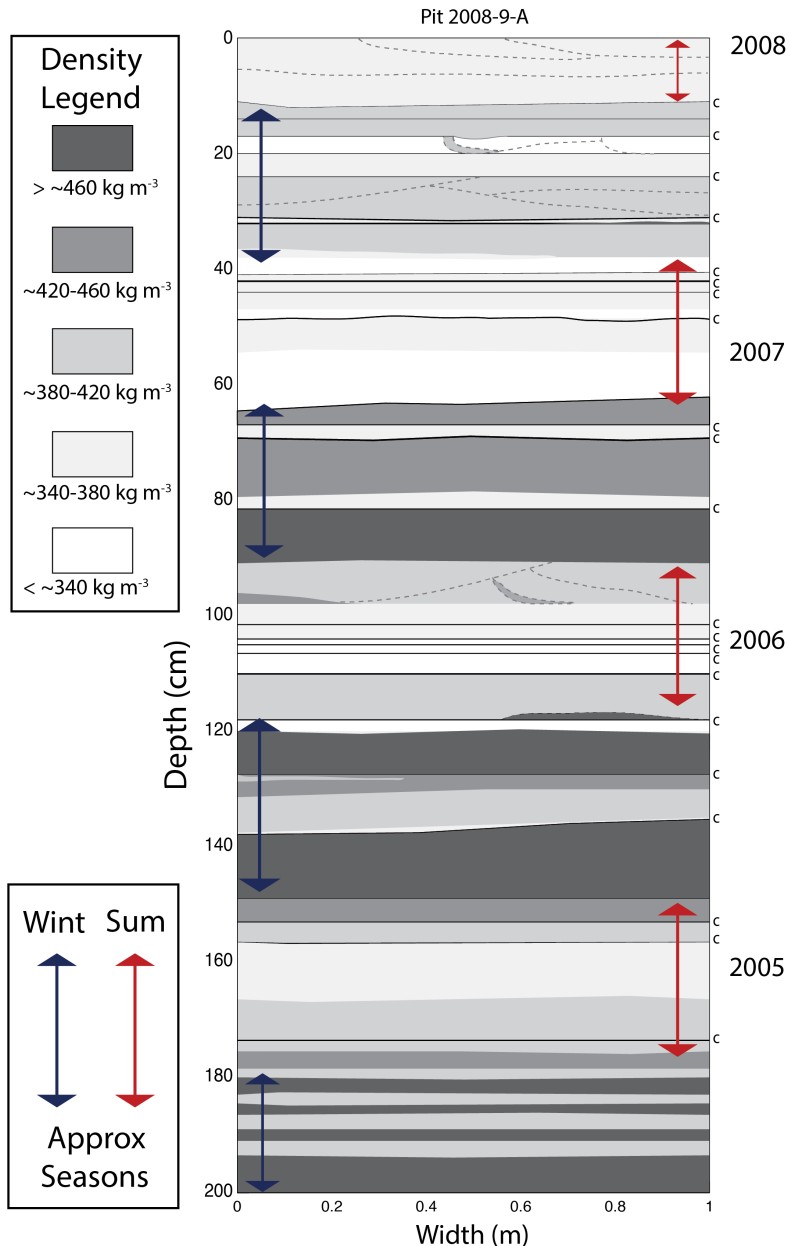

**Figure 13:** A detailed view of data for snow pit 2008-09-A, including wall map, density profile,
annual layer picks, and crusts occurrences. Density, layering, and feature preservation are again
noted as in Fig. 12. Individual crusts are identified with a labeled "c" along the vertical axis.
Seasonal accumulation layers "picked" visually in the pit (shown with red and blue arrows)..
These observations indicate a somewhat regular pattern of equally-distributed yearly
accumulation at WAIS Divide with clear annual signals.

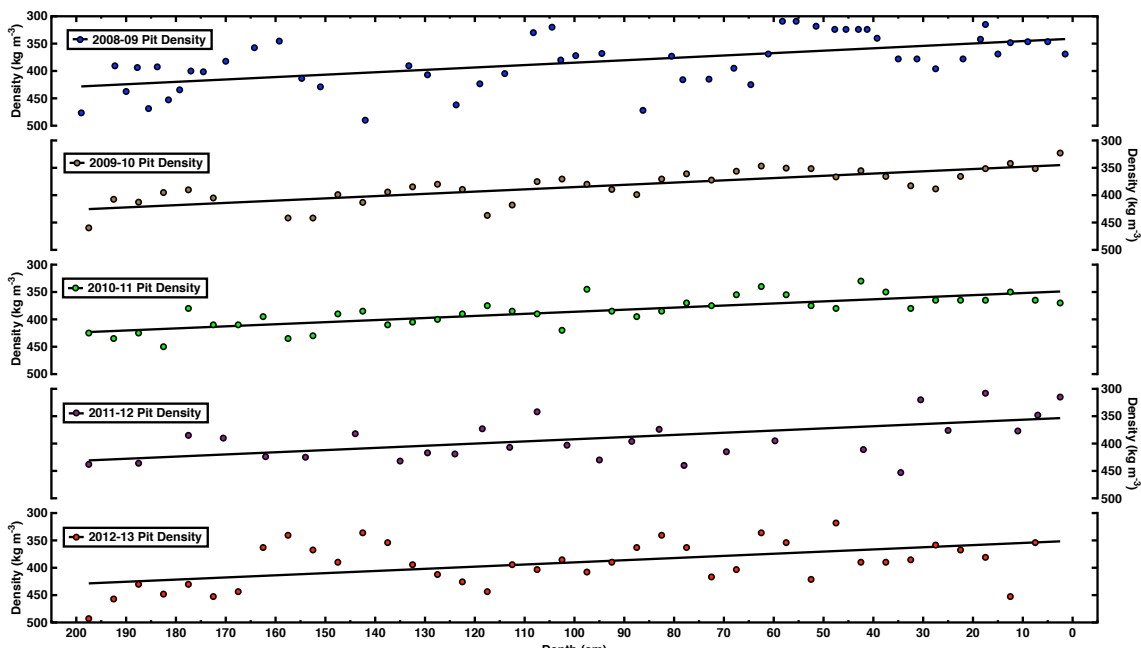

**Figure 14:** Density profiles measured in snow pits from five concurrent seasons at WAIS Divide
(2008-2012). Each pit showed a high degree of sample-to-sample variability as measured
densities were widely-spaced within the upper 2 meters of firn; estimated annual signals were still
identifiable, however. Measurements yielded an overall average density of 386.6 ± 3.2 kg m$^{-3}$ for
the upper 2 meters of firn across all 5 pits, with nearly identical linear trend-line slopes of ~0.4 kg
m$^{-3}$cm$^{-1}$ with depth.

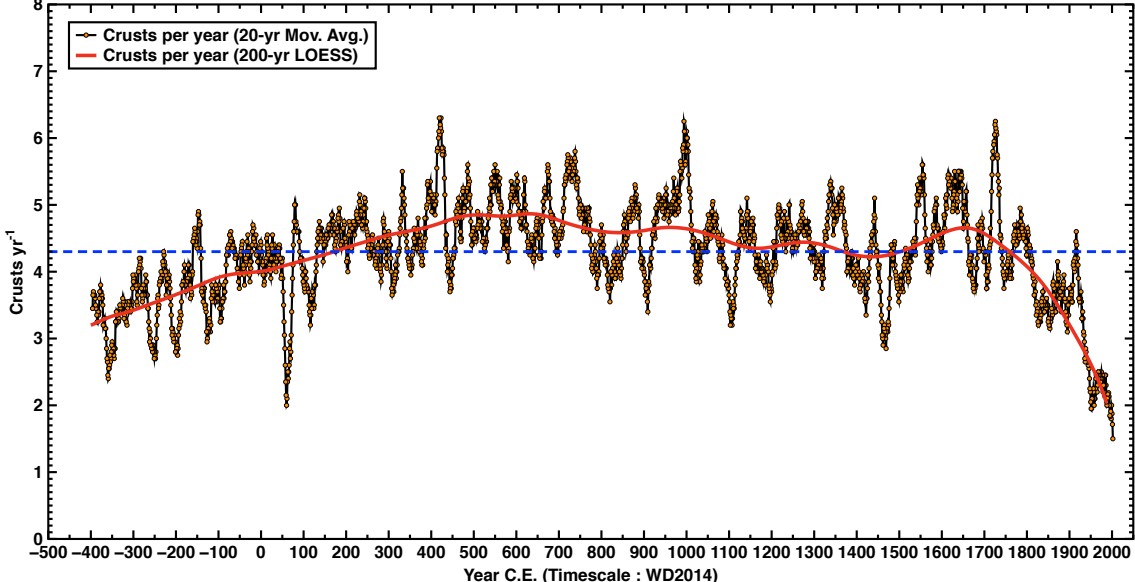


**Figure 15:** History of crust occurrence (crusts year[-1]) in the bubbly-ice zone of the WDC06A
core that we studied in detail (~120 – 577 m depth); ages (C.E.) are from the WD2014 depth-age
scale). 10,268 unique crusts were documented in the core, for an average rate of 4.3 ± 2 per year
(dashed blue line). Data are shown as 20-yr moving averages for ease of view, with an added 1[st]-
order LOESS smoothing trend-curve (200-yr bin-width). The sharp decline in crust prevalence
after ~1750 C.E. may be due to observational biasing in the shallow firn.

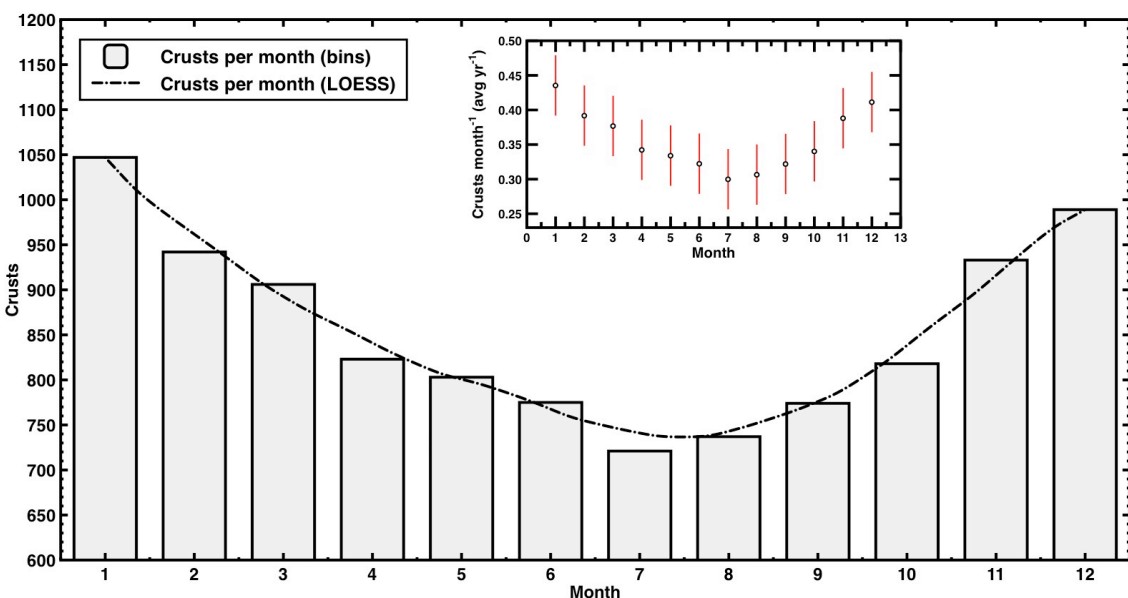

**Figure 16:** Crust distribution by month (1=January, 2=February,…12=December) based on
assumption that each summer pick in the WD2014 depth-age scale is January 1, and then
interpolating linearly. Crusts occur year-round but more commonly in summer accumulation. The
smoothed curve is a 1$^{st}$-order LOESS trend curve (width = 2). Data shown for 2400-yr record.
Inset shows average crusts per month (±1σ).

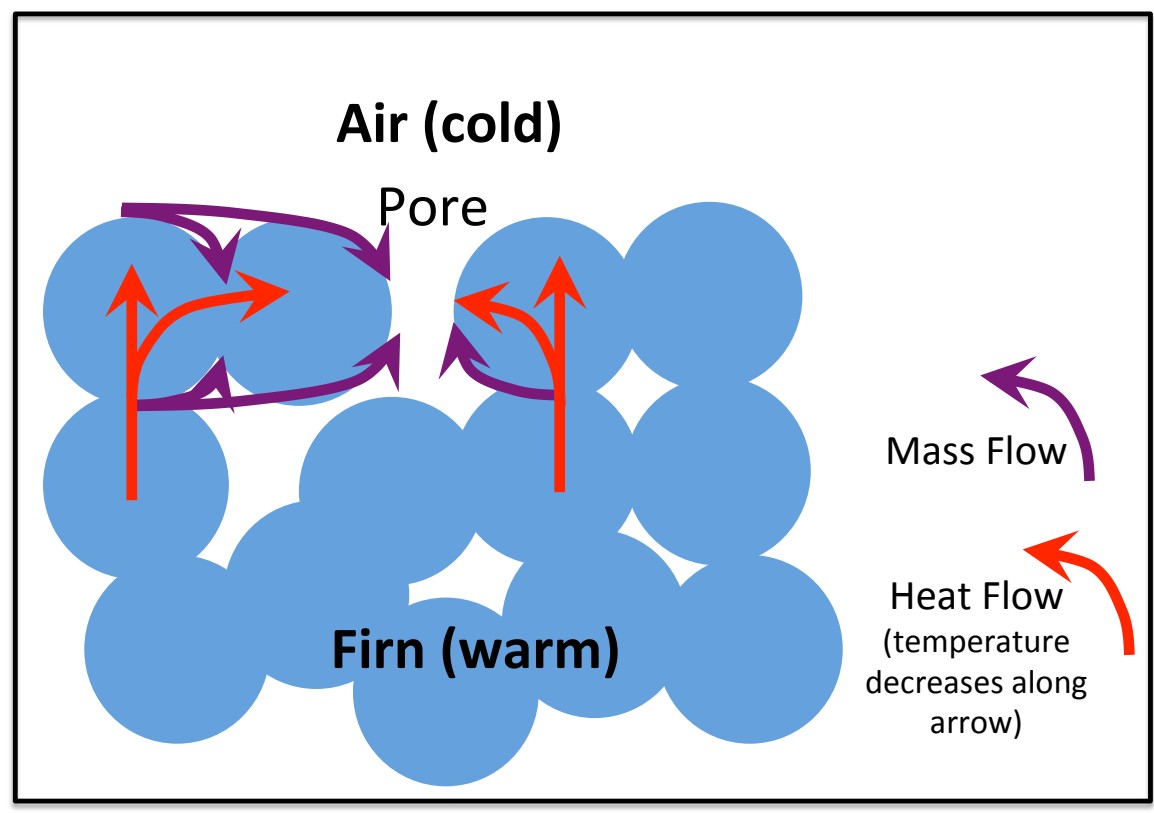

**Figure 17:** Schematic illustrating possible mass and heat transports during during formation of a
single-grain glazed crust, when the near-subsurface is warmer than the surface.  Heat flow is
primarily through the grain structure (blue), so pores (white) in the surface layer will be colder
than interconnected grains, favoring mass transport from the grains to those pores, increasing
density of the surface layer.

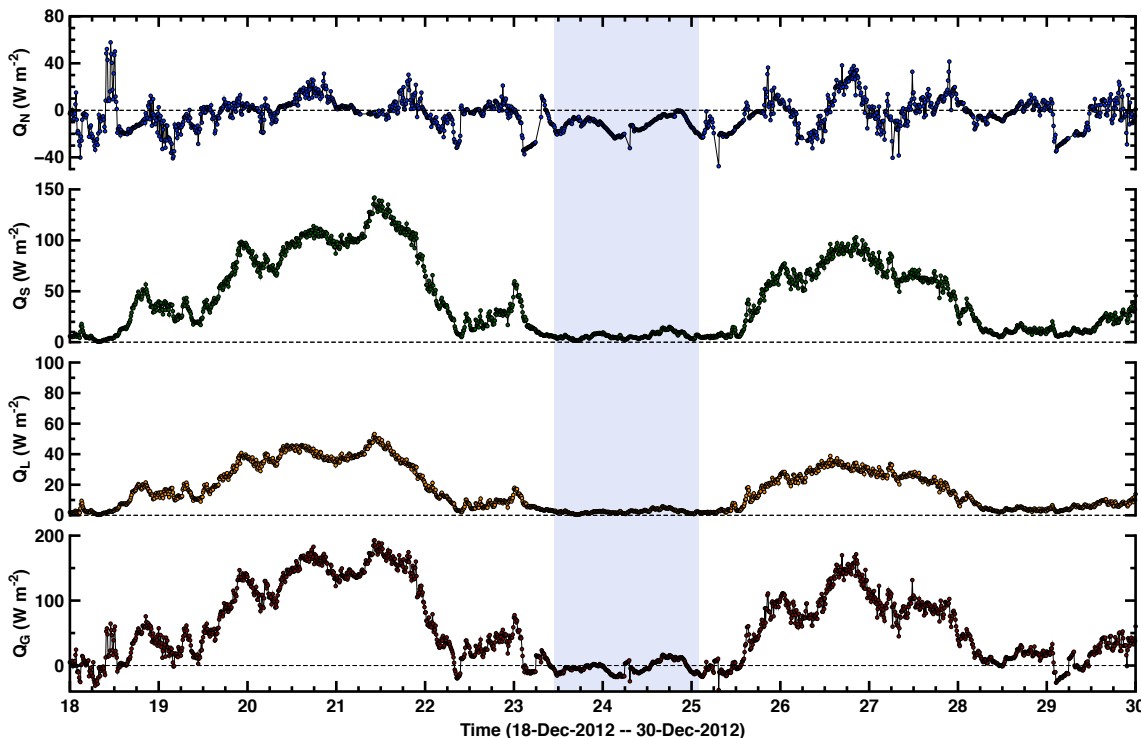

**Figure 18:** Surface energy budget over 12 days in 2012-13 season. Shading highlights the ~36-hr
period with episodes of glaze formation, polygonal cracking, and surface hoar formation (see also
Figure 8). Total net radiation ($Q_N$), turbulent sensible heat flux ($Q_S$), turbulent latent heat flux
($Q_L$) and calculated ground heat flux ($Q_G$) are shown. Dashed lines in all plots indicate zero
values. All dates and times are GMT (-12 WAIS local time).
**Table 1:** Field observation table (see also Figs. 5, 6, 7, 8).

| Field Season | Observation Window | Observation Duration | AWS | Other Instrumentation | Pit |
|---|---|---|---|---|---|
| 2008-2009[1] | 12-Dec-2008 : 10-Jan-2009 | ~29 days | -- | -- | x |
| 2009-2010[1] | 27-Dec-2009 : 25-Jan-2010 | ~29 days | W,H,T | -- | x |
| 2010-2011 | 20-Dec-2010 : 09-Jan-2011 | ~20 days | W,H,T | -- | x |
| 2011-2012[1] | 25-Dec-2011 : 04-Jan-2012 | ~12 days | W,H,T,I | Dual Li-Cor LI200 sensors Kipp-Zonen CNR2 sensor | x |
| 2012-2013[1] | 18-Dec-2012 : 30-Dec-2012 | ~12 days | W,H,T,I | Shallow PRD strings[2] | x |

W,H,T,I - Wind, Humidity, Temperature, Insolation
[1]Fegyveresi, 2015
[2]Muto et al., 2011
