# Peer review of "Surface formation, preservation, and history of low-porosity crusts at the WAIS Divide site, West Antarctica."

_The Cryosphere, 2016_

## Referee Comment (RC1) · M. Schneebeli (Referee) · 29 Jul 2016

The paper by Fegyveresi et al presents glaciological and meteorological observations concerning the formation of a thin crust at the snow surface around the WAIS drill site. The existence of this glazed crust has been described for several decades, but detailed information about the formation is lacking. This paper contributes to the understanding of the formation of these specific layers.

The methodology to investigate the formation of these layers is rather old-fashioned, and is not taking into account developments in snow characterization. The paper is in most aspects descriptive, with little quantitative information especially about the microstructure of the snow.

[Figure]

The paper describes in much detail the observations, but little quantitative data analysis and no modelling at all. The authors formulate extensively a general hypothesis on the formation of these layers, but do not substantiate their claims. The recent developments in snow metamorphism are not considered, and essential aspects in the interpretation are missing, especially concerning the radiation balance and the thermal conditions of the snowpack in the topmost layers of the snowpack.

I suggest the following revisions: - The short and longwave radiation balance should be calculated and used in the interpretation of the formation of the crusts. - Ideally, the weather data should to model the thermal conditions of the snowpack in the top 30 cm, this should not be a major difficulty (there are several snowpack models easily available, as Crocus and SNOWPACK). - A statistical analysis of the processes is needed, it does not become clear in the paper if the same weather conditions occur without formation of a glazed surface. - The hypotheses concerning the formation should be reformulated and quantified based on the results above

Based on the old-fashioned methods, several questions will probable remain open concerning the microstructural properties of the snow. It is a pity that now snow samples were cast using Diethyl-phthalte, this technique was already used in 1957 in Antarctica.

A general comment is also that the authors seem not to be aware of the difference between the terms "diagenesis" and "metamorphism". Diagenesis describes the densification by internal compaction or by a foreign sintering material. Metamorphism describes the recrystallization of minerals. Diagenetic processes in snow and firn occur in Antarctica below the isothermal zone (i.e. a few meters depth). Above, the dominant process is metamorphism.

Specific remarks: l 38 According to instruments, these are radiation sensors, not insulation sensors

l 40 There is no detailed data later in the manuscript about the crack spacing, so delete in abstract

l 42 If this theory is correct, then a very strong convective vapor transport would be necessary, and calculations (see e.g. Ebner et al, Calonne et al.)

l 48 Was this layering also found in the snowpit?

l 54 Are there any measurements done on the spatial extent of these layers (e.g. in the snowpit)?

l 110 The radiation instruments are described here in detail, but the data not used, why? These data are essential for the interpretation of crust formation. Especially snow surface temperature can determined precisely from the pyrgeometers.

l 165 I miss in this description the following information, resolved with mm-vertical resolution: density and specific surface area, ideally also coordination number. This information is available by a number of techniques, even with simple thick-sectioning of cast samples. Fig. 9 is just a photo with no quantitative information.

l 170 Is there any statistics on size? This is very descriptive.

l 201 How are the PRD-strings influenced by solar heating? The 10 cm depths is clearly not sufficient to prevent solar heating.

l 204 A 3 K over 40 cm results only in a temperature gradient of 7.5 K m-1, far too small to create a relevant mass flux (see e.g. Pinzer et al, 2012).

l 211 I would expect detailed drawings of the layer, how many snowpits measured, detailed statistics (spacing, size, ...)

l 217 How should I interpret "most commonly"? Any statistics here?

l 221 I think you have either to reduce the approximate sign (and the about 2 m snow pit, or with the over a depth of 2.000 m?). What was the number of pits?

l 240 "should be quite accurate", what do mean in numbers by this?

l 249 The upper 30 cm are definitively not firn but snow, see the great discussion in

Anderson, D. L., and C. S. Benson (1963), The densification and diagenesis of snow, in Ice and Snow: Properties, Processes and Applications, edited by W. D. Kingery, pp. 391–411, MIT Press.

l 257 -274 The following descriptions are not a discussion, but a narrative of the observations

l 275-282 I agree that surface hoar is an atmospheric deposition on the ground, but this paragraph disagrees with your hypothesis of vapor creeping out of cracks.

l 283 Measured density data for crusts (for any snow layers) is not presented in this manuscript

l 301 I could not find any calculated temperature gradients in the result section

Suggestions on recent papers on snow metamorphism

Calonne, N., F. Flin, C. Geindreau, B. Lesaffre, and S. Rolland du Roscoat (2014), Study of a temperature gradient metamorphism of snow from 3-D images: time evolution of microstructures, physical properties and their associated anisotropy, The Cryosphere, 8, 2255–2274, doi:10.5194/tc-8-2255-2014.

Ebner, P. P., C. Andreoli, M. Schneebeli, and A. Steinfeld (2015), Tomography-based characterization of ice-air interface dynamics of temperature gradient snow metamorphism under advective conditions, Journal of Geophysical Research: Earth Surface, 120(12), 2437–2451, doi:10.1002/2015JF003648.

Pinzer, B. R., M. Schneebeli, and T. U. Kaempfer (2012), Vapor flux and recrystallization during dry snow metamorphism under a steady temperature gradient as observed by time-lapse micro-tomography, The Cryosphere, 6(5), 1141–1155, doi:10.5194/tc-6-1141-2012.

---

## Referee Comment (RC2) · Anonymous Referee #2 · 1 May 2017

Review of

Surface formation, preservation, and history of low-porosity crusts at the WAIS Divide site, West Antarctica.

By John M. Fegyveresi, Richard B. Alley, Atsuhiro Muto, Anaïs J. Orsi and Matthew K. Spencer

General

This is a comparatively comprehensive study of the formation of so-called surface crusts, involving daily observations of surface crust formation at the WAIS divide site in West Antarctica over five consecutive summers (2008/09 to 2012/13), including annual

shallow snow-pit studies, snow temperature profiles and data (including shortwave radiation measurements) from an automatic weather station (AWS). The main conclusion is that crusts form most commonly in the summer from the effects of a large daily temperature cycle. There also appears to be crust formation in winter, as yet for unknown reasons.

The paper provides useful and original data for model development and evaluation, and the topic is suitable for publication in TC. The paper is rather descriptive, but useful, as the authors state in line 275: "Our data provide strong constraints on models of many of the observed processes." However, the value of the study and analysis would be greatly quantified if the AWS and snow temperature data were used to calculate the surface energy balance, see comments below. I recommend to do this, which will require major revisions.

Major comments

While explicit modelling of microphysical snow processes is beyond this MS's scope, a more quantitative interpretation can be achieved relatively easily by using the AWS and snow temperature data to close the surface energy balance. This will greatly aid the discussion by quantifying the sign and magnitude of surface energy fluxes, including the transport of water vapour by sublimation/deposition, during episodes of crust formation. See e.g. Van As and others (2005; 2006).

l. 45: "often warmed the near-surface snow above the air temperature, contributing to mass transfer..." This suggests that temperature gradient is a sufficient condition for sublimation, but this requires a specific humidity gradient (a less stringent condition).

l. 118: Were relative humidity measurements corrected for low-temperature offsets (See Andersen and others, 1994)?

l. 152: "accumulation at the site is relatively evenly distributed through the year, justifying this approximation"; this may be true for the climatological precipitation, but is there

quantitative support that this holds for individual years as well?

l. 201: "following the air 202 temperatures as expected". Figure 10: Surface energy balance considerations dictate that the amplitude of the daily cycle in surface temperature exceeds that in air temperature, to allow for nocturnal cooling and daytime heating by sensible heat exchange. This appears not to be the case in these time series. Please comment.

Figure 13: please translate y-axis into average crusts per individual month, and include standard deviation as error bar. Mention ice core time interval in caption.

Minor/Textual comments

l. 38: "insolation sensors" refers to incoming shortwave radiation. Better: "shortwave radiation sensors"

l. 113: pyrogeometers -> pyrgeometers

l. 191: crust removal -> hoar removal (?)

l. 330: "warm and windy air masses" an air mass cannot be windy, please reformulate.

References

Anderson, P., 1994: A method for rescaling humidity sensors at temperatures well below freezing, J. Atmos. Oceanic. Technol. 11, 138801391.

Van As, D. andÂăM. R. van den Broeke, 2006: Structure and dynamics of the summertime atmospheric boundary layer over the Antarctic plateau, II: Heat, momentum and moisture budgets,ÂăJournal of Geophysical ResearchÂă111, D007103, doi:10.1029/2005JD006956.

Van As, D.,ÂăM. R. van den Broeke, R. S. W. van de Wal, 2005: Daily cycle of the surface layer and energy balance on the high Antarctic plateau,ÂăAntarctic ScienceÂă17, 121-133.

---

## Author Comment (AC1) · 25 Jul 2017

Review of Surface formation, preservation, and history of low-porosity crusts at the WAIS Divide site, West Antarctica. By John M. Fegyveresi, Richard B. Alley, Atsuhiro Muto, Anaïs J. Orsi and Matthew K. Spencer

[Figure]

General This is a comparatively comprehensive study of the formation of so-called surface crusts, involving daily observations of surface crust formation at the WAIS divide site in West Antarctica over five consecutive summers (2008/09 to 2012/13), including annual shallow snow-pit studies, snow temperature profiles and data (including shortwave ra- diation measurements) from an automatic weather station (AWS). The main conclusion is that crusts form most commonly in the summer from the effects of a large daily tem- perature cycle. There also appears to be crust formation in winter, as yet for unknown reasons. The paper provides useful and original data for model development and evaluation, and the topic is suitable for publication in TC. The paper is rather descriptive, but useful, as the authors state in line 275: "Our data provide strong constraints on models of many of the observed processes." However, the value of the study and analysis would be greatly quantified if the AWS and snow temperature data were used to calculate the surface energy balance, see comments below. I recommend to do this, which will require major revisions.

"We did correct all noted issues and responded to reviewer comments, specifically with the inclusion of more on the crust extent and on related snow pit studies (that were previously left out), although we did not add either additional modeling or a full energy balance study. The paper is already quite long, and we have specific challenges with attempting to incorporate either of these additional and lengthy studies. We are aware of the additional papers cited by the referees, and note the large amount of careful work involved. We do also hope to be able investigate further the possible effects of solar heating on the specific type of PRT sensors. We have added relevant citations to our paper. With our manuscript being this long and the difficulties with adding such large/expansive analyses, we believe it is better to write a phenomenological paper first and then address modeling and a full energy balance in a separate/future study."

Major comments While explicit modelling of microphysical snow processes is beyond this MS's scope, a more quantitative interpretation can be achieved relatively easily by using the AWS and snow temperature data to close the surface energy balance.
This will greatly aid the discussion by quantifying the sign and magnitude of surface energy fluxes, including the transport of water vapour by sublimation/deposition, during episodes of crust formation. See e.g. Van As and others (2005; 2006).

l. 45: "often warmed the near-surface snow above the air temperature, contributing to mass transfer. . ." This suggests that temperature gradient is a sufficient condition for sublimation, but this requires a specific humidity gradient (a less stringent condition. The relative humidity in the air may be below saturation; that in the snow is likely to be much closer to saturation because of proximity to the moisture source in the snow. So temperature gradient really is enough.

"We reworded for clarity."

l. 118: Were relative humidity measurements corrected for low-temperature offsets (See Andersen and others, 1994)?

"They were. All humidity values shown are corrected and represented in terms of saturation vapor pressure over ice (as described by Anderson 1994)"

l. 152: "accumulation at the site is relatively evenly distributed through the year, justifying this approximation"; this may be true for the climatological precipitation, but is there quantitative support that this holds for individual years as well?

"We added back the more-detailed pit study (including 3 figures) to help better illustrate/quantify this."

l. 201: "following the air temperatures as expected". Figure 10: Surface energy balance considerations dictate that the amplitude of the daily cycle in surface temperature exceeds that in air temperature, to allow for nocturnal cooling and daytime heating by sensible heat exchange. This appears not to be the case in these time series. Please comment.

"It is accurate that if sensible heat transfer is occurring, the temperature must be as the reviewer states; however there is no guarantee that sensible heat transfer is occurring.

We edited wording and identified specific sensors for clarity."

Figure 13: please translate y-axis into average crusts per individual month, and include standard deviation as error bar. Mention ice core time interval in caption.

"Adjusted figure to show an inset showing average crusts per month with 1 sigma stdev."

Minor/Textual comments l. 38: "insolation sensors" refers to incoming shortwave radiation. Better: "shortwave radiation sensors" . "Insolation sensors" was used as there were both short and longwave sensors used. . .depending on the year.

"We adjusted text to "short and longwave radiation sensors"."

l. 113: pyrogeometers -> pyrgeometers .

"Corrected"

l. 191: crust removal -> hoar removal (?)

"Correct. Adjusted text to read "hoar"."

l. 330: "warm and windy air masses" an air mass cannot be windy, please reformulate.

"Reworded to "Such warm air masses paired with these high winds,""

References Anderson, P., 1994: A method for rescaling humidity sensors at temperatures well below freezing, J. Atmos. Oceanic. Technol. 11, 138801391. (Added) Van As, D. andÂa ÌĘM. R. van den Broeke, 2006: Structure and dynamics of the summertime atmospheric boundary layer over the Antarctic plateau, II: Heat, mo- mentum and moisture budgets,Âa ÌĘJournal of Geophysical ResearchÂa ÌĘ111, D007103, doi:10.1029/2005JD006956. VanAs,D.,Âa ÌĘM.R.vandenBroeke,R.S.W.vandeWal,2005:Dailycycleofthesur- face layer and energy balance on the high Antarctic plateau,Âa ÌĘAntarctic ScienceÂa ÌĘ17, 121-133.

---

## Author Comment (AC2) · 26 Jul 2017

Fegyveresi et al. M. Schneebeli (Referee) schneebeli@slf.ch

The paper by Fegyveresi et al presents glaciological and meteorological observations concerning the formation of a thin crust at the snow surface around the WAIS drill site. The existence of this glazed crust has been described for several decades, but detailed information about the formation is lacking. This paper contributes to the understanding of the formation of these specific layers.

[Figure]

The methodology to investigate the formation of these layers is rather old-fashioned, and is not taking into account developments in snow characterization. The paper is in most aspects descriptive, with little quantitative information especially about the microstructure of the snow.

The paper describes in much detail the observations, but little quantitative data analysis and no modelling at all. The authors formulate extensively a general hypothesis on the formation of these layers, but do not substantiate their claims. The recent developments in snow metamorphism are not considered, and essential aspects in the interpretation are missing, especially concerning the radiation balance and the thermal conditions of the snowpack in the topmost layers of the snowpack.

I suggest the following revisions: - The short and longwave radiation balance should be calculated and used in the interpretation of the formation of the crusts. - Ideally, the weather data should to model the thermal conditions of the snowpack in the top 30 cm, this should not be a major difficulty (there are several snowpack models easily available, as Crocus and SNOWPACK). - A statistical analysis of the processes is needed, it does not become clear in the paper if the same weather conditions occur without formation of a glazed surface. - The hypotheses concerning the formation should be reformulated and quantified based on the results above

Based on the old-fashioned methods, several questions will probable remain open concerning the microstructural properties of the snow. It is a pity that now snow samples were cast using Diethyl-phthalte, this technique was already used in 1957 in Antarctica. A general comment is also that the authors seem not to be aware of the difference between the terms "diagenesis" and "metamorphism". Diagenesis describes the densification by internal compaction or by a foreign sintering material. Metamorphism describes the recrystallization of minerals. Diagenetic processes in snow and firn occur in Antarctica below the isothermal zone (i.e. a few meters depth). Above, the dominant process is metamorphism. –

"Replaced terms to reflect metamorphism rather than diagenesis."

"We did correct all noted issues and responded to reviewer comments, specifically with the inclusion of more on the crust extent and on related snow pit studies (that were previously left out), although we did not add either additional modeling or a full energy balance study. The paper is already quite long, and we have specific challenges with attempting to incorporate either of these additional and lengthy studies. We are aware of the additional papers cited by the referees, and note the large amount of careful work involved. We do also hope to be able investigate further the possible effects of solar heating on the specific type of PRT sensors. We have added relevant citations to our paper. With our manuscript being this long and the difficulties with adding such large/expansive analyses, we believe it is better to write a phenomenological paper first and then address modeling and a full energy balance in a separate/future study."

Specific remarks: l 38 According to instruments, these are radiation sensors, not insulation sensors

"Reworded to "short and longwave radiation sensors""

l 40 There is no detailed data later in the manuscript about the crack spacing, so delete in abstract

Deleted

l 42 If this theory is correct, then a very strong convective vapor transport would be necessary, and calculations (see e.g. Ebner et al, Calonne et al.)

"Don't have a sufficient calculation here. We cannot partition accurately the relative contributions of vapor transport from below versus condensation of vapor in the atmosphere from other sources, but the wording 'may have contributed' should make this clear."

l 48 Was this layering also found in the snowpit?

"It was and the detailed snow-pit investigation was added back to manuscript including text and 3 additional figures."

l 54 Are there any measurements done on the spatial extent of these layers (e.g. in the snowpit)?

"We have semi-quantitative measures, but not accurate maps, and we have additional calculations from occurrence of partial crusts in the core, which are well-quantified; a reference to that latter calculation can be found in the later-referenced PhD (Fegyveresi, 2015)."

l 110 The radiation instruments are described here in detail, but the data not used, why? These data are essential for the interpretation of crust formation. Especially snow surface temperature can determined precisely from the pyrgeometers.

"They are used in the figures to show trends, but not used in more-detailed calculations. The final instrumentation was not the same from year-to-year; only in the final year was the full net radiometer installed. . .thus so a multi-year comparison of identical radiation data could not be performed."

l 165 I miss in this description the following information, resolved with mm-vertical resolution: density and specific surface area, ideally also coordination number. This information is available by a number of techniques, even with simple thick-sectioning of cast samples.

"Edited for clarity using snowpit data (See other new figures)."

l 170 Is there any statistics on size? This is very descriptive.

"No additional measured field statistics exist, however we added more detail for clarity. Also, related thesis chapter does include field observation data table with some additional observation notes on surface glazes (See Fegyveresi, 2015)"

l 201 How are the PRD-strings influenced by solar heating? The 10 cm depths is clearly

not sufficient to prevent solar heating.

"As originally designed (see Muto et al., 2011), the sensor itself is housed inside a small aluminum tube, encased in glue...which minimizes direct solar heating. Still, we recognize that the very top surface sensors in this study (S0) that's placed under just a very small layer of snow is likely to be influenced slightly by some fraction of solar heating. Specific calibrations for this effect were not carried out in the field, but we felt the influence was likely minimal due to the nature of the sensor design and the "surface sensors" not being directly exposed to solar radiation."

l 204 A 3 K over 40 cm results only in a temperature gradient of 7.5 K m-1, far too small to create a relevant mass flux (see e.g. Pinzer et al, 2012).

Gradients steepen towards surface (See Alley et al., 1990 ; eg. Their figure 2). A 3k over 40 cm gradient likely has much steeper gradient near the surface...sufficient to drive the relevant mass flux.

l 211 I would expect detailed drawings of the layer, how many snowpits measured, detailed statistics (spacing, size, ...) Added back in the detailed pit maps and associated figures.

l 217 How should I interpret "most commonly"? Any statistics here?

"As stated previously, 45% greater occurrence in Summers. Edited for clarity"

l 221 I think you have either to reduce the approximate sign (and the about 2 m snow pit, or with the over a depth of 2.000 m?). What was the number of pits?

Stated previously – one snowpit per year for a 5 year period (so 5 pits)...but text has been added back in to clarify. Also, to clarify, all pits were measured to 2 meters, but due to sampling spacing and 5 cm thickness, the bottom sample was centered on 197.5 cm total depth (covering 195-200 cm).

l 240 "should be quite accurate", what do mean in numbers by this?

Reworded for clarity. . ."are well-constrained"

l 249 The upper 30 cm are definitively not firn but snow, see the great discussion in Anderson, D. L., and C. S. Benson (1963), The densification and diagenesis of snow, in Ice and Snow: Properties, Processes and Applications, edited by W. D. Kingery, pp. 391–411, MIT Press.

Reworded to indicate that the upper portion of firn is generally considered snow.

l 257 -274 The following descriptions are not a discussion, but a narrative of the observations

Relabled the section "Synopsis and Discussion"

l 275-282 I agree that surface hoar is an atmospheric deposition on the ground, but this paragraph disagrees with your hypothesis of vapor creeping out of cracks.

The paragraph states that there are two sources of hoar growth. . .from above and below. While some hoar growth was clearly related to high humidity fog episodes, the most dominant process related to hoar growth was sublimation related due to vapor transport.

l 283 Measured density data for crusts (for any snow layers) is not presented in this manuscript

Added in all snowpit density data and related figures. Also added text indicating all crusts densities measured and/or estimated over 400 kg m-3 .

l 301 I could not find any calculated temperature gradients in the result section Would have to be related to modeleling. . .need to calculate.

Referenced back to appropriate figures/data

Suggestions on recent papers on snow metamorphism Calonne, N., F. Flin, C. Geindreau, B. Lesaffre, and S. Rolland du Roscoat (2014), Study of a temperature gradient metamorphism of snow from 3-D images: time evo- lution of microstructures, physical properties and their associated anisotropy, The Cryosphere, 8, 2255–2274, doi:10.5194/tc-8-2255-2014. Ebner, P. P., C. Andreoli, M. Schneebeli, and A. Steinfeld (2015), Tomography-based characterization of ice-air interface dynamics of temperature gradient snow metamor- phism under advective conditions, Journal of Geophysical Research: Earth Surface, 120(12), 2437–2451, doi:10.1002/2015JF003648. Pinzer, B. R., M. Schneebeli, and T. U. Kaempfer (2012), Vapor flux and recrystallization during dry snow metamorphism under a steady temperature gradient as observed by time-lapse micro-tomography, The Cryosphere, 6(5), 1141–1155, doi:10.5194/tc-6-1141-2012.
* * *

---

## Author Response (AR3)

M. Schneebeli (Referee)

schneebeli@slf.ch

**Suggestions for revision or reasons for rejection (will be published if the paper is accepted for final publication)**

**"firn" and "snow" are now used inconsistently in the manuscript. Please define the term. In my opinion this is always snow, as (1) density is below 550 kg m-3 (2) temperature gradient metamorphism dominant (3) the age of the porous ice is irrelavant for the physics.**

We have corrected all instances of "firn" to "snow" where appropriate based upon reviewers suggestion. Two instances were left as "Firn" as it was appropriate based on discussion of deeper sections within the firn. (see lines 77 and 270)

**The surface energy budget (lines 363 ff) and Fig. 18 seem to contain a major error: turbulent and latent heat fluxes seem to be too high by a factor 100. Please check.**

We did find an error in the calculation of the turbulent fluxes (based on an incorrect use of the roughness parameter) and we thank the reviewer for noting this. This has been corrected in the text and figure. Our overall result for DeltaT calculation changed only slightly however, as the time period of interest was during a low wind episode (and therefore low turbulent fluxes). See text for updated results. In addition, we've reworded the text slightly in this newer section for clarity (~lines 370 – 420).

**Note to Editor**: We have moved the corrected Figure 18 to the supplemental document (Supplemental Fig. 7). While related to the new text and energy balance study, based on the nature of the calculation, we didn't feel it was absolutely necessary to include in the primary text. We do still feel it's worth keeping for reference however, and thus moved it to the supplemental doc. If this is unacceptable, please let us know and we will move it back.

**Note to Editor:** References have been updated and DOI numbers added/corrected.